# Large-scale phenotypic drug screen identifies neuroprotectants in zebrafish and mouse models of retinitis pigmentosa

Liyun Zhang[1], Conan Chen[1†], Jie Fu[2], Brendan Lilley[1], Cynthia Berlinicke[1], Baranda Hansen[1‡], Ding Ding[3], Guohua Wang[1§], Tao Wang[2,4,5], Daniel Shou[2], Ying Ye[2], Timothy Mulligan[1], Kevin Emmerich[1,6], Meera T Saxena[1], Kelsi R Hall[7#], Abigail V Sharrock[3,7], Carlene Brandon[8], Hyejin Park[9], Tae-In Kam[9,10], Valina L Dawson[9,10,11,12], Ted M Dawson[9,10,11,12], Joong Sup Shim[13], Justin Hanes[1,2], Hongkai Ji[3], Jun O Liu[11,14], Jiang Qian[1], David F Ackerley[7], Baerbel Rohrer[8], Donald J Zack[1,2,6,12,15], Jeff S Mumm[1,2,6,12]*

[1]Department of Ophthalmology, Wilmer Eye Institute, Johns Hopkins University, Baltimore, United States; [2]The Center for Nanomedicine, Wilmer Eye Institute, Johns Hopkins University, Baltimore, United States; [3]Department of Biostatistics, Johns Hopkins University, Baltimore, United States; [4]School of Chemistry, Xuzhou College of Industrial Technology, Xuzhou, China; [5]College of Light Industry and Food Engineering, Nanjing Forestry University, Nanjing, China; [6]Department of Genetic Medicine, Johns Hopkins University, Baltimore, United States; [7]School of Biological Sciences, Victoria University of Wellington, Wellington, New Zealand; [8]Department of Ophthalmology, Medical University of South Carolina, Charleston, United States; [9]Department of Neurology, Johns Hopkins University, Baltimore, United States; [10]Institute for Cell Engineering, Johns Hopkins University, Baltimore, United States; [11]Department of Pharmacology and Molecular Sciences, Johns Hopkins University, Baltimore, United States; [12]Solomon H. Snyder Department of Neuroscience, Johns Hopkins University, Baltimore, United States; [13]Faculty of Health Sciences, University of Macau, Taipa, Macau, China; [14]Department of Oncology, Johns Hopkins University, Baltimore, United States; [15]Department of Molecular Biology and Genetics, Johns Hopkins University, Baltimore, United States

*For correspondence: jmumm3@jhmi.edu

Present address: †University of Colorado Anschutz Medical Campus, Aurora, United States; ‡St. Jude Children's Research Hospital, Memphis, United States; § School of Computer Science and Technology, Harbin Institute of Technology, Harbin, China; #Faculty of Chemistry, Norwegian University of Life Sciences, Bergen, Norway

**Abstract** Retinitis pigmentosa (RP) and associated inherited retinal diseases (IRDs) are caused by rod photoreceptor degeneration, necessitating therapeutics promoting rod photoreceptor survival. To address this, we tested compounds for neuroprotective effects in multiple zebrafish and mouse RP models, reasoning drugs effective across species and/or independent of disease mutation may translate better clinically. We first performed a large-scale phenotypic drug screen for compounds promoting rod cell survival in a larval zebrafish model of inducible RP. We tested 2934 compounds, mostly human-approved drugs, across six concentrations, resulting in 113 compounds being identified as hits. Secondary tests of 42 high-priority hits confirmed eleven lead candidates. Leads were then evaluated in a series of mouse RP models in an effort to identify compounds effective across species and RP models, that is, potential pan-disease therapeutics. Nine of 11 leads exhibited neuroprotective effects in mouse primary photoreceptor cultures, and three promoted photoreceptor survival in mouse *rd1* retinal explants. Both shared and complementary mechanisms of action were implicated across leads. Shared target tests implicated *parp1*-dependent cell death in our zebrafish RP model. Complementation tests revealed enhanced and additive/synergistic neuroprotective effects of paired drug combinations in mouse photoreceptor cultures and

zebrafish, respectively. These results highlight the value of cross-species/multi-model phenotypic drug discovery and suggest combinatorial drug therapies may provide enhanced therapeutic benefits for RP patients.

## Introduction

Inherited retinal diseases (IRDs) encompass a group of genetically linked retinopathies characterized by progressive photoreceptor death (*Duncan et al., 2018*). IRDs lead to irreversible vision loss, for which treatment strategies are limited. Retinitis pigmentosa (RP), the most common IRD with approximately 1.5–2.5 million RP patients a worldwide (*Dias et al., 2018*; *Hartong et al., 2006*; *Verbakel et al., 2018*), is characterized by early onset night blindness, gradual loss of visual field, and eventual loss of central vision (*Ferrari et al., 2011*; *Hamel, 2006*). The initial pathological feature of RP is selective rod photoreceptor cell death causing night blindness, which is then followed by loss of cone photoreceptors and eventual full blindness (*Léveillard et al., 2014*). Mutations in more than 70 genes have been linked to RP (*Dias et al., 2018*; https://sph.uth.edu/retnet/). How these mutations affect gene function or initiate aberrant photoreceptor cell loss is largely unknown. Given the genetic diversity, pan-disease therapeutics are highly desirable. Accordingly, the purpose of our study was to identify compounds capable of promoting rod cell survival in multiple RP models and across species.

As RP progression is relatively protracted, and even small numbers of surviving rod photoreceptors can preserve cone photoreceptor function (*Guadagni et al., 2015*; *Hartong et al., 2006*; *Punzo et al., 2012*), pharmacological interventions aimed at slowing rod photoreceptor death are sought (*Duncan et al., 2018*; *Wubben et al., 2019*). However, currently there are no effective therapies for promoting rod photoreceptor survival. As a means of discovering new pharmacological treatments, target-directed high-throughput screening (HTS) approaches have been highly successful in identifying compounds that bind to and/or modulate disease-implicated molecules. However, many promising leads fail during late-stage animal model testing or clinical trials (*Munos, 2009*; *Sams-Dodd, 2013*; *Scannell et al., 2012*). This trend has renewed interest in phenotypic drug discovery (PDD), a complementary approach where drug effects are evaluated in cells or living disease models (*Bickle, 2010*; *Lee et al., 2012*; *Swinney, 2013*). A number of first-in-class drugs were recently discovered using PDD (*Eder et al., 2014*; *Swinney and Anthony, 2011*; *Swinney, 2013*). To expand opportunity on this front, we developed a PDD platform enabling quantitative HTS (qHTS; *Inglese et al., 2006*) in zebrafish (*Walker et al., 2012*; *Wang et al., 2015a*; *White et al., 2016*).

Zebrafish offer several distinct advantages as a retinal disease modeling system (*Angueyra and Kindt, 2018*; *Richardson et al., 2017*). First, the structure of the zebrafish retina is similar to the other vertebrates (*Angueyra and Kindt, 2018*; *Richardson et al., 2017*; *Schmitt and Dowling, 1999*). In particular, the zebrafish retina is 'cone rich' like the human retina. Second, about 70% of human genes have at least one ortholog in zebrafish (*Howe et al., 2013*). Moreover, all RP-associated genes listed in RetNet (https://sph.uth.edu/retnet/) have conserved zebrafish orthologs. Third, the zebrafish retinal system develops quickly being fairly mature by day five of development (*Brockerhoff et al., 1995*; *Moyano et al., 2013*; *Schmitt and Dowling, 1999*). Fourth, zebrafish are amendable to large-scale chemical screening due to their high fecundity rate, small size, and ease of visualizing and quantifying a variety of phenotypes (*Mathias et al., 2012*; *Zon and Peterson, 2005*). To streamline such screens, we developed a high-throughput plate reader-based method for quantifying reporter gene expression in vivo ('ARQiv'; *Walker et al., 2012*). Recently, we adapted the ARQiv system to human stem-cell-derived retinal organoids (*Vergara et al., 2017*) to enable cross-species PDD. To realize full throughput potential, ARQiv was combined with robotics-based automation to create 'ARQiv-HTS' (*Wang et al., 2015b*; *White et al., 2016*).

Here, to identify neuroprotective compounds promoting rod photoreceptor survival, ARQiv-HTS was used to perform a large-scale chemical screen in a transgenic zebrafish model enabling inducible rod photoreceptor cell death (*Walker et al., 2012*; *White et al., 2017*). In these fish, YFP-NTR (a yellow fluorescent protein-nitroreductase bacterial enzyme fusion protein) is selectively expressed in rod photoreceptors. NTR expression enables inducible ablation of rod cells upon exposure to prodrug substrates, such as metronidazole (Mtz), which are thought to cause apoptosis through

**eLife digest** Photoreceptors are the cells responsible for vision. They are part of the retina: the light-sensing tissue at the back of the eye. They come in two types: rods and cones. Rods specialise in night vision, while cones specialise in daytime colour vision. The death of these cells can cause a disease, called retinitis pigmentosa, that leads to vision loss. Symptoms often start in childhood with a gradual loss of night vision. Later on, loss of cone photoreceptors can lead to total blindness. Unfortunately, there are no treatments available that protect photoreceptor cells from dying.

Research has identified drugs that can protect photoreceptors in animal models, but these drugs have failed in humans. The classic way to look for new treatments is to find drugs that target molecules implicated in a disease, and then test them to see if they are effective. Unfortunately, many drugs identified in this way fail in later stages of testing, either because they are ineffective, or because they have unacceptable side effects. One way to reverse this trend is to first test whether a drug is effective at curing a disease in animals, and later determining what it does at a molecular level. This could reveal whether drugs can protect photoreceptors before research to discover their molecular targets begins. Tests like this across different species could maximise the chances of finding a drug that works in humans, because if a drug works in several species, it is more likely to have shared target molecules across species.

Applying this reasoning, Zhang et al. tested around 3,000 drug candidates for treating retinitis pigmentosa in a strain of zebrafish that undergoes photoreceptor degeneration similar to the human disease. Most of these drug candidates already have approval for use in humans, meaning that if they were found to be effective for treating retinitis pigmentosa, they could be fast-tracked for use in people.

Zhang et al. found three compounds that helped photoreceptors survive both in zebrafish and in retinas grown in the laboratory derived from a mouse strain with degeneration similar to retinitis pigmentosa. Tests to find out how these three compounds worked at the molecular level revealed that they interfered with a protein that can trigger cell death. The tests also found other promising compounds, many of which offered increased protection when combined in pairs.

Worldwide there are between 1.5 and 2.5 million people with retinitis pigmentosa. With this disease, loss of vision happens slowly, so identifying drugs that could slow or stop the process could help many people. These results suggest that placing animal testing earlier in the drug discovery process could complement traditional target-based methods. The compounds identified here, and the information about how they work, could expand potential treatment research. The next step in this research is to test whether the drugs identified by Zhang et al. protect mammals other than mice from the degeneration seen in retinitis pigmentosa.

activated metabolites that cause DNA damage (*Curado et al., 2007*; *White and Mumm, 2013*). Close to 3000 largely human-approved drugs were tested across six concentrations (i.e. using qHTS principles, *Inglese et al., 2006*) in more than 350,000 zebrafish larvae. Statistically, 113 hits were identified as hits and 42 of the top performing compounds advanced through confirmation and orthogonal assays. Eleven compounds passed all secondary tests and moved forward as lead drug candidates. Validation tests in an orthogonal series of mouse RP models followed, reasoning that drugs effective across species and/or independent of disease mutation may translate better clinically. All leads were screened in primary mouse photoreceptor cell cultures and a subset of leads was evaluated in retinal explants from the retinal degeneration 1 (*Pde6b^{rd1}*, hereafter *rd1*) mutant mouse model of RP. Finally, one of three top-performing leads was tested in retinal degeneration 10 mutant retinas in vivo (*Pde6b^{rd10}*, hereafter *rd10*).

An analysis of potential mechanisms of action (MOA) suggested both shared and complementary targets/pathways. The role of Poly (ADP-ribose) polymerase (PARP), the most common shared target, was evaluated using chemical inhibition, genetic targeting, and western blot analysis. The results implicated PARP as a key mediator of NTR/Mtz-mediated rod cell ablation, and PARP1 inhibition as a shared MOA for a subset of leads. As PARP-dependent cell death pathways are implicated in the etiology of Parkinson's disease (*Kam et al., 2018*), RP (*Power et al., 2020*), and other neurodegenerative disorders (*Fan et al., 2017*; *Fatokun et al., 2014*), NTR/Mtz-mediated cell ablation

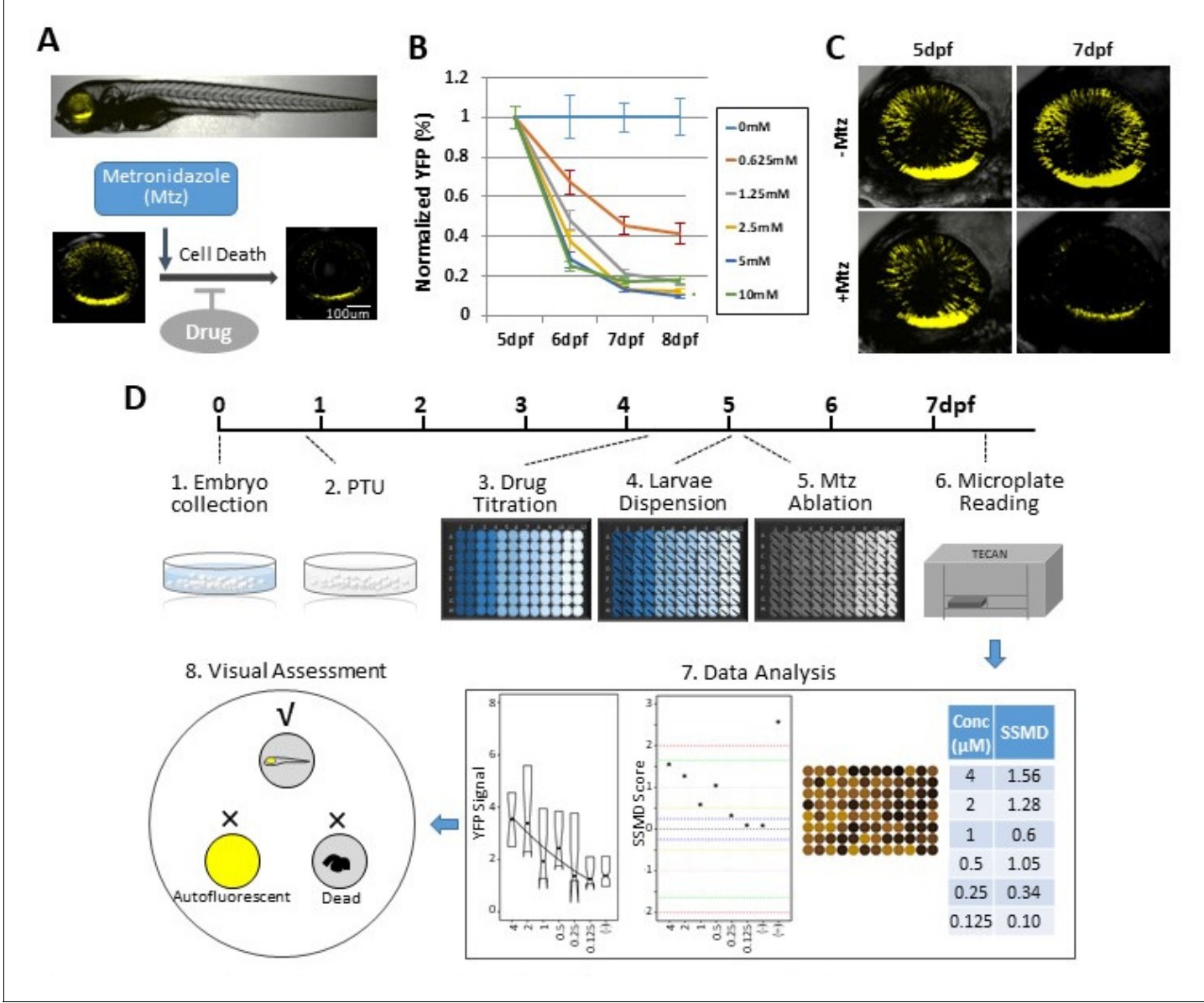

**Figure 1.** Zebrafish-inducible RP model and schematic of large-scale phenotypic screen. (**A–C**) Prodrug (Metronidazole, Mtz) inducible ablation of rod photoreceptors in *rho:YFP-NTR* larvae. (**A**) In vivo confocal images of a *rho:YFP-NTR* larvae showing transgene expression specificity and whole retinas before and after Mtz-induced ablation of rod photoreceptors; schematic shows proposed neuroprotective effect of screened compounds. (**B**) Optimization of Mtz treatment regimen for large-scale screen: at five dpf, *rho:YFP-NTR* larvae were exposed to 10, 5, 2.5, 1.25, or 0.625 mM Mtz and YFP levels assessed daily by fluorescence microplate reader from 5 to 8 dpf (sample size: 56 larvae per group, two experimental repeats). The average YFP signal (±sem) for each Mtz treatment group is plotted as the percentage relative to non-ablated (0 mM Mtz) control signals per day. The 10, 5, and 2.5 mM Mtz-treated groups produced minimal YFP signal intensities (<20%) at 7 dpf (*Supplementary file 1a*; *Figure 1B—source data 1*). A 2.5 mM Mtz treatment over two days (5–7 dpf), the minimal Mtz concentration achieving maximal ablation, was therefore chosen for the large-scale primary screen. (**C**) Time series in vivo confocal images of representative non-ablated (-Mtz control, upper panel) and 2.5 mM Mtz-treated (+Mtz control, lower panel) retinas at 5 dpf (pre-Mtz) and 7 dpf (post-Mtz). By 7 dpf, only a limited number of YFP-positive cells are detectable in the +Mtz retina, mainly concentrated in a ventral band of high rod cell density. (**D**) Schematic of primary drug screening process: (1) At 0 dpf, large numbers of embryos were collected. (2) At 16 hpf, PTU was added to suppress melanization. (3) At 4 dpf, individual drugs were dispensed and titrated in 96-well plates using robotic liquid handlers; 16 wells per concentration (two columns) and six concentrations per drug. (4) At 5 dpf, the COPAS was used to dispense individual larvae into single wells of the drug titration 96-well plates. (5) After a 4 hr pre-exposure to drugs, larvae were treated with 2.5 mM Mtz to induce rod cell ablation. (6) At 7 dpf, YFP signals were quantified by fluorescence microplate reader assay. (7) Same day data analysis using a custom R code (https://github.com/mummlab/ARQiv2; *Ding and Zhang, 2021*) was used to plot signal to background ratios, SSMD plot, microplate heat map, and SSMD score table. (8) Drug plates producing SSMD scores of ≥1 were visually inspected using fluorescence stereomicroscopy to exclude autofluorescent and lethal compounds.

*Figure 1 continued on next page*

*Figure 1 continued*

The online version of this article includes the following source data and figure supplement(s) for figure 1:

**Source data 1.** Mtz titration test.

**Figure supplement 1.** Immunohistological labeling of rod and cone photoreceptors in *rho:YFP-NTR* larvae.

may serve as an inducible and titratable methodology for modeling neurodegenerative disease. To interrogate complementary MOA, lead compounds were tested in pairs in mouse photoreceptor cell culture assays and in zebrafish. Intriguingly, enhanced survival effects were common in mouse photo-receptor cell cultures, while additive and even synergistic effects were evident in zebrafish for the majority of pairs tested. Taken together, our results suggest drugs providing neuroprotective effects across diverse RP models, between species, and/or through complementary MOA, will provide promising new therapeutic opportunities for IRD/RP patients.

## Results

### Establishing a large-scale reporter-assisted neuroprotectant screen in zebrafish larvae

We initiated our study using a robotics-automated large-scale in vivo screening system to identify compounds that promoted rod photoreceptor survival in a transgenic zebrafish line enabling targeted ablation of rod photoreceptors, *Tg(rho:YFP-Eco.NfsB)gmc500*, hereafter, *rho:YFP-NTR* (*Walker et al., 2012*; *White et al., 2017*). In this line, a 3.7 kb *rhodopsin* (*rho*) promoter fragment (*Hamaoka et al., 2002*) drives transgene expression exclusively in rod photoreceptor cells (*Figure 1A*). The transgene is a fusion protein linking a yellow fluorescent protein (YFP) reporter to a nitroreductase prodrug converting enzyme (NTR, encoded by the *nfsB* gene from *Escherichia coli*). NTR expression enables pro-drug inducible targeted cell ablation (*Curado et al., 2007*; *Pisharath et al., 2007*). Exposing *rho:YFP-NTR* fish to the prodrug metronidazole (Mtz) leads to the selective death of rod photoreceptors and concomitant loss of YFP (*Figure 1A–C*), physiologically mimicking the onset of RP (*Hamel, 2006*). An immunohistological analysis of rod and cone photoreceptor markers was performed on 7 days post-fertilization (dpf) zebrafish retinal sections to test if Mtz-induced ablation was specific to rod cells. In non-ablated controls, rod outer segment labeling was well correlated with YFP expression (*Figure 1—figure supplement 1A,B*; -Mtz, arrows). In Mtz-treated retinas, rod outer segment labeling and YFP expression were markedly diminished (*Figure 1—figure supplement 1A,B*; +Mtz). Cone photoreceptor labeling showed no overlap with YFP expression, and cone cell labeling appeared similar in non-ablated and Mtz-treated retinas (*Figure 1—figure supplement 1C*), suggesting cone cells were not affected by Mtz exposure or by acute rod cell loss.

Prior studies suggest NTR/Mtz-mediated ablation occurs by apoptosis (*Chang et al., 1993*; *Doonan et al., 2003*; *Portera-Cailliau et al., 1994*; *Zeiss et al., 2004*) and loss of rods in RP has also been linked to apoptosis (*Chang et al., 1993*; *Doonan et al., 2003*; *Portera-Cailliau et al., 1994*; *Zeiss et al., 2004*). Thus, we reasoned *rho:YFP-NTR* fish could be used to identify compounds that protected rods from apoptosis. To identify neuroprotective compounds with this model (i.e. drugs that sustain YFP expression after Mtz exposure), we used our established plate reader-based ARQiv-HTS assay (*Figure 1D*; *Walker et al., 2012*; *White et al., 2016*).

We first determined optimal conditions for inducing rod cell loss while maintaining larval health in a 96-well plate format. Major aspects of retinal cytogenesis are largely complete by five dpf in zebra-fish (*Schmitt and Dowling, 1999*; *Stenkamp, 2011*). Reporter expression in *rho:YFP-NTR* larvae has also stabilized by this time point (*Unal Eroglu et al., 2018*), consistent with *rho* expression (*Raymond et al., 1995*). We therefore chose 5 dpf to initiate Mtz-induced rod cell ablation. We previously determined that rod cell loss reached a nadir at 7 dpf following a 24 hr pulse of 10 mM Mtz at 5 dpf (*Walker et al., 2012*). We reasoned that a 48 hr Mtz exposure initiating at 5 dpf would maximize the signal window to test for neuroprotective effects. Concluding the experiment by 7 dpf also avoids challenges associated with feeding, as zebrafish can subsist on their yolk sac up to that time point (*Hernandez et al., 2018*; *Jardine and Litvak, 2003*). However, 10 mM Mtz treatments

extending beyond 24 hr become increasingly toxic (*Mathias et al., 2014*) and removing Mtz from microtiter plates after a 24 hr pulse could not be easily automated. We therefore sought a 48 hr Mtz treatment regimen sufficient for inducing maximal rod cell loss by 7 dpf that showed no evidence of general toxicity.

Five concentrations of Mtz were tested across a twofold dilution series from 10 mM to 625 µM. YFP reporter signals were quantified daily from 5 to 8 dpf using ARQiv. Changes in YFP levels were calculated as percentages normalized to non-ablated YFP controls. The data showed concentration-dependent reductions in YFP with maximal loss observed at 7 dpf (*Figure 1B*). Mtz exposures at 0.625, 1.25, 2.5, 5, and 10 mM led to a 55, 79, 87, 87, and 83% decrease in YFP detection, respectively (*Supplementary file 1a*). Although no lethality was observed for any condition, signs of distress were evident for 10 mM Mtz exposures (i.e. reduced motility). At $\leq$5 mM Mtz, however, no signs of stress were observed. As 2.5 mM Mtz was the lowest concentration producing maximal loss of YFP detection, and confocal imaging verified YFP loss following a 2-day (5–7 dpf) 2.5 mM Mtz treatment (*Figure 1C*), this condition was selected as the treatment regimen for the large-scale screen.

Previously established power analysis methods using ablated and non-ablated controls (*White et al., 2016*) determined that a sample size of nine larvae was sufficient to detect a 50% neuroprotective effect. For ease of dispensing, microtiter plate formatting, and to account for larval dispensing errors, we increased the sample size to 16 larvae per condition for the primary screen. We also reasoned that this would also allow us to detect subtler neuroprotective effects. The strictly standardized mean difference quality control (SSMD QC) score was 1.67, indicating the assay was of sufficient quality to justify a large-scale screening effort (*Zhang, 2011*).

To establish a positive control, we tested 17 compounds and one compound 'cocktail' previously implicated as neuroprotectants in mammalian RP models (*Supplementary file 2a*). Unfortunately, none sustained YFP expression at the concentrations tested (4 µM to 125 nM). However, none have proven effective in RP patients either, suggesting a lack of conservation of neuroprotective effects across species. Our screen was designed to address this issue by testing for neuroprotective effects of these compounds across species and between different RP models. Fortunately, a compound identified as a retinal cell neuroprotectant by the Zack lab (manuscript in preparation) did show-dose-dependent effects on YFP levels. This compound was therefore used as a positive control (POS) for assay performance throughout the large-scale screen.

## Primary screen

The Johns Hopkins Drug Library (JHDL) was chosen for the large-scale screen. The JHDL is comprised of nearly 3000 compounds, most being human-approved drugs (*Shim and Liu, 2014*). To minimize false discovery rates, all compounds were tested using qHTS principles (*Inglese et al., 2006*) – that is, across six concentrations (4 µM - 125 nM) using a twofold dilution series. The screen largely followed published ARQiv-HTS methodologies (*Wang et al., 2015b*; *White et al., 2016*), with additional assay-specific details (*Figure 1D*, steps 1–8). In all, 2934 compounds were screened and more than 350,000 transgenic zebrafish larvae evaluated. Real-time data analysis was performed as previously detailed (*White et al., 2016*) to generate: (1) a plot of YFP signal levels, (2) a plot of SSMD scores across all tested concentrations, (3) a signal intensity heat map of each plate, and (4) an SSMD score table (*Figure 1D*, step 7). Compounds producing SSMD scores of $\geq$one were considered potential hits and flagged for visual inspection to assess fluorescence and general morphology using a stereo fluorescence microscope. This step facilitated elimination of false-positive compounds producing aberrant fluorescence due to larval toxicity (six drugs) or compound autofluorescence (27 drugs; *Figure 1C*, step 8; *Supplementary file 2b*). Additionally, this allowed visual confirmation of sustained YFP expression within the retina. At the conclusion of the primary screen, 113 compounds were identified as hits (*Supplementary file 2c*). Hit compounds were classified according to the highest SSMD score achieved across all concentrations tested, and whether concentration-dependent effects were observed. SSMD scores suggested one drug produced a strong effect (SSMD of 2–3); three had semi-strong effects (1.645–2), 21 showed moderate effects (1.28–1.645), and 88 had fairly moderate effects (1–1.28) (*Supplementary file 2c*). Forty-two drugs showed concentration-dependent effects, while 71 exhibited discontinuous or singular concentration effects. 'On label' mechanism of action (MOA) information available for the 113 hits implicated more than 50 targets/pathways in rod cell neuroprotection (*Supplementary file 2d*).

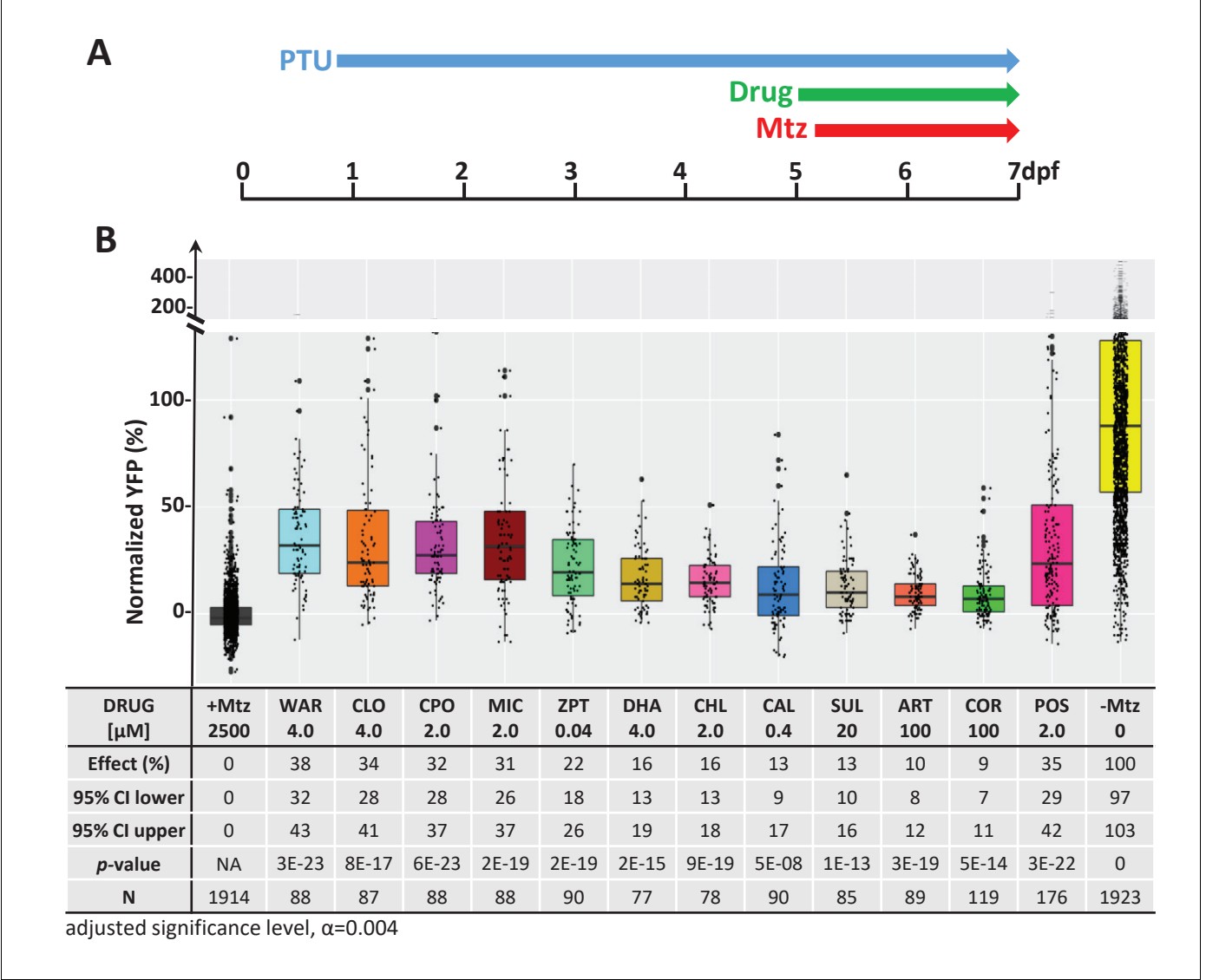

Figure 2. Confirmation of neuroprotective effects of lead compounds in *rho:YFP-NTR* zebrafish larvae. (**A**) Diagram of confirmation assay protocol (see *Figure 1D* for further details). (**B**) Box plots of rod cell survival effects of eleven confirmed lead compounds (arrayed by level of neuroprotection) and the positive control compound (POS). YFP signal intensities were normalized to the signal window defined by setting ablated controls (+Mtz) at 0% and non-ablated controls (-Mtz) at 100% to account for: (1) individual variation in reporter expression between fish and, (2) fluctuations in signal window across assays, thus allowing data from identical conditions to be pooled across experimental repeats. Survival effects (normalized YFP %), 95% confidence intervals, p-values, and sample sizes (N) for each condition are shown in the table below. Student's *t*-test was used to calculate p-values for each condition relative non-ablated controls (0 mM Mtz). Bonferroni correction for multiple comparisons resulted in an adjusted significance level of 0.004 (α=0.004). A minimum of three experimental replicates was performed for each condition (*Figure 2—source data 1*). No statistical differences in larval survival were observed for lead compounds relative to their respective +Mtz controls, except for DHA (86%) and CHL (87%; Fisher's exact test, p<0.05). Common names, IUPAC names, abbreviations, PubChem CID, and chemical structures of lead compounds are listed in *Table 1*. Lead compound abbreviations: WAR, Warfarin; CLO, Cloxyquin; CPO, Ciclopirox olamine; MIC, Miconazole; ZPT, Zinc pyrithione; DHA, Dihydroartemisinin; CHL, Chloroxine; CAL, Calcimycin; SUL, Sulindac; ART, Artemisinin; COR, Cortexolone; POS, positive control. Other abbreviations: CI, confidence interval; Mtz, Metronidazole; PTU, N-phenylthiourea.

The online version of this article includes the following source data and figure supplement(s) for figure 2:

**Source data 1.** Lead compound confirmation.
**Figure supplement 1.** Test of lead compound effects on NTR enzymatic activity.
**Figure supplement 1—source data 1.** Compound effects on NTR activity.
**Figure supplement 2.** Test of lead compound effects on rod cell neogenesis during development.
**Figure supplement 2—source data 1.** Compound effects on rod cell neogenesis.
**Figure supplement 3.** Test of lead compound effects on rod cell regeneration.

*Figure 2 continued on next page*

**Table 1.** Eleven confirmed lead compounds.

Common names, IUPAC names, abbreviations, PubChem CID, and chemical structures of confirmed lead compounds listed in order of efficacy.

| Common and IUPAC names | Abbr. | PubChem CID | Structure |
|---|---|---|---|
| Warfarin<br>4-hydroxy-3-(3-oxo-1-phenylbutyl) chromen-2-one | WAR | 54678486 | |
| Cloxyquin 5-Chloro-8-hydroxyquinoline | CLO | 2817 | |
| Ciclopirox olamine<br>2-aminoethanol;<br>6-cyclohexyl-1-hydroxy-4-methylpyridin-2-one | CPO | 38911 | |
| Miconazole<br>1-[2-(2,4-dichlorophenyl)—2-[(2,4-dichlorophenyl)methoxy]ethyl]imidazole | MIC | 4189 | |
| Zinc pyrithione zinc;<br>1-oxidopyridin-1-ium-2-thiolate | ZPT | 26041 | |
| Dihydroartemisinin<br>(4S,5R,9R,10S,12R,13R)—1,5,9-trimethyl-11,14,15,16-tetraoxatetracyclo [10.3.1.0$^{4,13}$.0$^{8,13}$] hexadecan-10-ol | DHA | 456410 | |
| Chloroxine<br>5,7-dichloroquinolin-8-ol | CHL | 2722 | |
| Calcimycin<br>5-(methylamino)—2-[[(2S,3R,5R,8S,9S)—3,5,9-trimethyl-2-[1-oxo-1-(1H-pyrrol-2-yl)propan-2-yl]—1,7-dioxaspiro[5.5]undecan-8-yl]methyl]—1,3-benzoxazole-4-carboxylic acid | CAL | 40486 | |
| Sulindac<br>2-[(3Z)—6-fluoro-2-methyl-3-[(4-methylsulfinylphenyl)methylidene]inden-1-yl] acetic acid | SUL | 1548887 | |

*Table 1 continued on next page*

*Table 1 continued*

| Common and IUPAC names | Abbr. | PubChem CID | Structure |
|---|---|---|---|
| Artemesinin (1*R*,4*S*,5*R*,8*S*,9*R*,12*S*,13*R*)—1,5,9-trimethyl-11,14,15, 16-tetraoxatetracyclo [10.3.1.0$^{4,13}$.0$^{8,13}$]hexadecan-10-one | ART | 68827 | |
| Cortexolone (8*R*,9*S*,10*R*,13*S*, 14*S*,17*R*)—17-hydroxy-17- (2-hydroxyacetyl)—10,13-dimethyl- 2,6,7,8,9,11,12,14,15,16-decahydro- 1*H*-cyclopenta[a]phenanthren-3-one | COR | 440707 | |

## Confirmation test

We next performed a series of confirmatory and orthogonal assays to evaluate a subset of 42 hit compounds prioritized by SSMD score, dose-response profile, and/or implicated MOA (*Supplementary file 2c*, highlighted compounds). Having extensive MOA data is a key advantage of testing human-approved compounds which we leveraged in a previous large-scale zebrafish PDD screen (*Wang et al., 2015b*). Similarly here, as studies have suggested inflammation plays a key role in retinal degeneration and regeneration (*Hollyfield et al., 2008*; *Mitchell et al., 2018*; *White et al., 2017*; *Yoshida et al., 2013*), hits implicated as modulators of inflammatory signaling were included as part of the prioritization scheme. In addition, several compounds that did not produce concentration-dependent effects were selected to test whether this criterion was useful in predicting reproducibility. All compounds were obtained from new sources to ensure reagent authenticity. To confirm survival promoting effects, three repeats of the primary screening assay (*Figure 2A*) were conducted, but using a wider concentration range to account for differences in reagent quality (from 100 μM to 1.28 nM, fivefold dilution series). If toxicity was observed at higher concentrations, dilution series were initiated at 10 μM or 1 μM. Using this strategy, 11 of the 42 prioritized hit compounds were confirmed as lead candidates (26%). Estimated effects on rod cell survival – that is, YFP signal levels relative to +Mtz/0.1% DMSO controls – ranged from 38 to 9% (*Figure 2B*; lead drug candidate names, abbreviations, PubChem CID, and chemical structures are provided in *Table 1*).

We next asked whether there was a correlation between SSMD scores and/or concentration-dependent effects and confirmation rates. Among 19 selected hit compounds with higher SSMD scores (≥1.3), seven (37%) were confirmed; among 23 with lower SSMD scores (1–1.28), four (17%) were confirmed. Of 27 selected hit compounds with a concentration-dependent trend, eight (30%) were confirmed. Conversely, of 15 compounds that did not show a concentration-dependent trend, 3 (20%) were confirmed. Among the 11 confirmed leads, 8 (73%) showed dose-dependent effects and 7 (64%) had higher SSMD scores (>1.28). These results suggest that prioritizing hit compounds by both relative SSMD scores and dose-dependent trends provides predictive value for confirming activity, consistent with qHTS principles (*Inglese et al., 2006*).

## Zebrafish model validation I: NTR inhibition

As rod cell death is induced by NTR-mediated reduction of the prodrug Mtz in our model, it is possible that some lead compounds simply suppressed NTR enzymatic activity. To test this, NTR activity was evaluated in the presence of each lead compound by assaying the reduction kinetics of the prodrug CB1954 in vitro (*Prosser et al., 2010*). To ensure any potential for NTR inhibition was accounted for, all compounds were tested at 300 μM (except MIC and CHL which precipitated at higher concentrations and were tested at 50 μM) that is, ~100-fold greater than neuroprotective concentrations. Compounds were deemed potential inhibitors if NTR activity was less than 75% of the NTR alone control. Seven compounds showed no evidence of NTR inhibition by this criterion, but four did: warfarin (WAR), ciclopirox olamine (CPO), calcimycin (CAL), and sulindac (SUL; *Figure 2— figure supplement 1A,B*). However, IC$_{50}$ measures ranged from 150 μM (for CPO) to 350 μM (for SUL, *Figure 2—figure supplement 1B*), approximately one to two orders of magnitude higher than

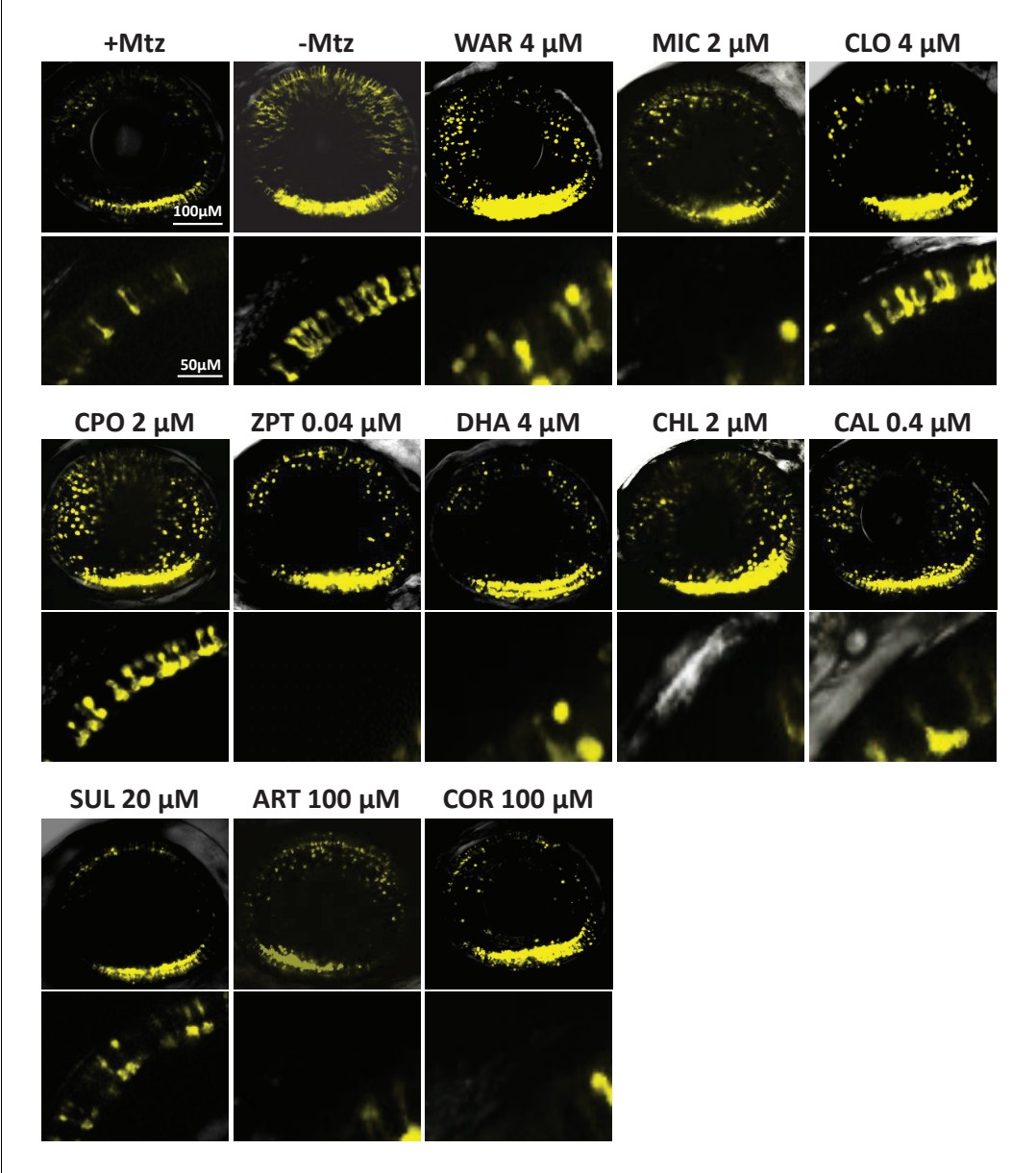

**Figure 3.** Confocal imaging of neuroprotective effects of confirmed lead drug compounds. Representative 7 dpf in vivo confocal images of YFP-expressing rod cells in an ablated control (+Mtz), non-ablated control (-Mtz), and in Mtz-exposed retinas treated with lead compounds from 5 to 7 dpf. Whole retina 3D image stacks (~150 micron depth) show loss or preservation of YFP-expressing rod cells throughout the retina, zoomed images of a more limited stack in the dorsal-nasal quadrant (~12 micron depth) provide rod cell morphological detail. Lead compound abbreviations: WAR, Warfarin; CLO, Cloxyquin; CPO, Ciclopirox olamine; MIC, Miconazole; ZPT, Zinc pyrithione; DHA, Dihydroartemisinin; CHL, Chloroxine; CAL, Calcimycin; SUL, Sulindac; ART, Artemisinin; COR, Cortexolone; POS, positive control. Other abbreviations: CI, confidence interval; Mtz, Metronidazole; PTU, N-phenylthiourea.

The online version of this article includes the following source data and figure supplement(s) for figure 3:

**Figure supplement 1.** Quantification of YFP intensity in *Figure 3* confocal images.

**Figure supplement 1—source data 1.** YFP quantification - intensity.

**Figure supplement 2.** Lead compound effects on outer segment morphology.

**Figure supplement 2—source data 1.** YFP quantification - volume per cell.

observed neuroprotective concentrations (i.e. 0.4–20 µM). When NTR reduction of Mtz was tested for WAR, CPO, and CAL at 300 µM, similarly weak inhibitory effects were observed (*Figure 2—figure supplement 1B*). The differences in concentrations between neuroprotective and NTR inhibitory activities diminish the likelihood that these leads act directly on NTR. To test this further, lead compounds were assayed for neuroprotective effects in NTR-independent mouse RP models (see below).

## Zebrafish model validation II: rod photoreceptor development

To control for the possibility that lead candidates promoted rod photoreceptor development, rather than neuroprotection, YFP levels were quantified in *rho:YFP-NTR* larvae exposed solely to lead drugs from 5 to 7 dpf (*Figure 2—figure supplement 2A*). Retinoic acid (RA, 1.25 µM) was used as a positive control as it promotes rod fates during development in zebrafish (*Hyatt et al., 1996*). RA-treated fish displayed statistically significant increases in YFP levels compared to untreated controls (*Figure 2—figure supplement 2B*). In contrast, none of the retinas treated with lead compounds exhibited increased YFP expression, suggesting they do not promote rod photoreceptor cell fate. In fact, three lead compounds cloxyquin (CLO), cortexolone (COR) and CPO reduced YFP signals relative to the untreated control (*Figure 2—figure supplement 2B*, asterisks), suggesting negative effects on rod cell development.

## Zebrafish model validation III: Regeneration

It is well known that the zebrafish retina regenerates (*Gorsuch and Hyde, 2014*; *Lenkowski and Raymond, 2014*; *Wan and Goldman, 2016*). Therefore, to determine whether lead compounds acted by stimulating regeneration, we used a previously described ARQiv assay designed to detect changes in rod cell replacement kinetics (*Walker et al., 2012*; *White et al., 2017*). Briefly, *rho:YFP-NTR* larvae were first treated with 10 mM Mtz at 5 dpf for 24 hr to induce rod cell loss. At 6 dpf, Mtz was washed out and larvae were treated with lead compounds at concentrations corresponding to maximal neuroprotective effects and YFP levels quantified at 9 dpf (*Figure 2—figure supplement 3A*). Dexamethasone, which accelerates rod cell regeneration kinetics (*White et al., 2017*), was used as a positive control. The results showed that none of the compounds increased rod cell regeneration rates (*Figure 2—figure supplement 3B*), while four compounds, dihydroartemisinin (DHA), CLO, CPO and MIC inhibited regeneration. These data suggest that lead compounds do not increase YFP levels by promoting rod cell regeneration.

## Zebrafish model validation IV: Confocal intravital microscopy

To test if lead compounds simply increased YFP signal intensity, rather than promoted rod cell survival, intravital microscopy was used to image Mtz-treated retinas ±lead compounds and non-ablated (-Mtz) controls. Mtz-treated retinas had reduced numbers of YFP-NTR-expressing rods (*Figure 3*, +Mtz). Conversely, rods in control retinas displayed robust YFP signal throughout the retina and elongated morphologies suggestive of healthy outer segments (*Figure 3*, -Mtz). In retinas exposed to Mtz and lead compounds, YFP signal loss was attenuated and rods typically displayed elongated morphologies (*Figure 3*). However, for some compounds, cells appeared rounded, suggesting degeneration was not fully inhibited (e.g. miconazole, MIC; *Figure 3*). To confirm lead compounds increased rod cell survival, confocal stacks of YFP-expressing rods were 3D-rendered and fluorescence volume quantified using Imaris software-based automated image analysis (*White et al., 2017*). The data showed a statistically significant increase in YFP volumes for all drug-treated groups (*Figure 3—figure supplement 1*), confirming that lead compounds promoted increased rod cell numbers and/or preserved outer segment morphology.

To further investigate if lead compounds preserved rod outer segments, we used a newly developed zebrafish transgenic line, *Tg(rho:GAP-YFP-2A-nfsB_Vv F70A/F108Y)jh405* (hereafter *rho:YFP-NTR2.0*). In this line, rods co-express membrane-tagged YFP and an improved NTR ('NTR 2.0'). The membrane-tagged YFP facilitates improved imaging of rod outer segments, while NTR 2.0 enables cell ablation at a reduced concentration of Mtz (e.g. 10 µM versus 2.5 mM; *Sharrock et al., 2020*). To assess if lead compounds preserved outer segment morphologies, intravital confocal imaging was performed as above but using the *rho:YFP-NTR2.0* line and 10 µM Mtz treatments. Qualitative image analysis suggested the majority, six of seven tested lead compounds, preserved rod outer

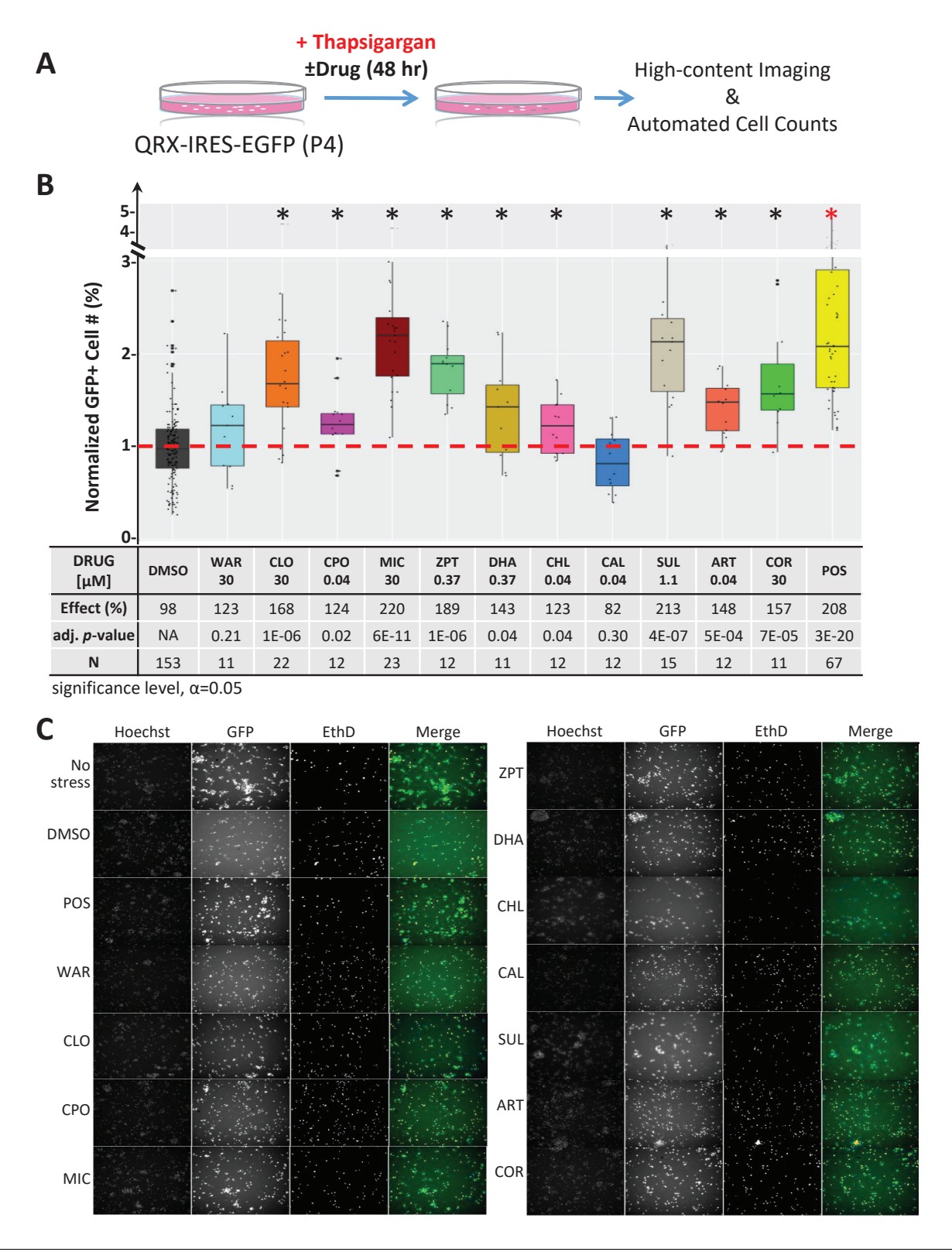

**Figure 4.** Lead effects pn mouse primary photoreceptor cells - thapsigargin-induced cell death. (**A**) Primary photoreceptor survival assay protocol. (**B**) Box plots of photoreceptor survival effects of lead compounds and the positive control compound (POS). Statistically significant survival effects are marked with an asterisk. Survival effects, adjusted p-values, and sample sizes (**N**) for each condition are shown in the table below. Adjusted p-values were calculated by performing Mann-Whitney *U* tests followed by false discovery rate (FDR) correction for multiple comparisons. Each assay consisted

*Figure 4 continued on next page*

*Figure 4 continued*

of six biological replicates per condition and a minimum of two experimental repeats was performed across all conditions (*Figure 4—source data 1*). (C) Representative images from each condition. Lead compound abbreviations: as in prior figures. Others: EthD, Ethidium homodimer; GFP, green fluorescent protein.

The online version of this article includes the following source data and figure supplement(s) for figure 4:

**Source data 1.** Compound tests of mouse photoreceptor culture - thapsigargin assay.
**Figure supplement 1.** Lead effects on mouse photoreceptor cultures - tunicamycin-induced cell death.
**Figure supplement 1—source data 1.** Quantification of mouse photoreceptor assay - tunicamycin-induced cell death.
**Figure supplement 2.** Paired lead compound survival effects in mouse photoreceptor cultures.
**Figure supplement 2—source data 1.** Paired drug tests - mouse photoreceptor cultures.

segments (*Figure 3—figure supplement 2A*). Reasoning that preservation of outer segments would equate to increased rod cell size, relative to rounded morphologies of dead or stressed cells, average rod cell sizes were calculated as the total YFP volume divided by the number of rod cells per each 3D confocal stack (i.e. average YFP volume per cell). Compared to rod ablated controls, five of seven lead compounds tested promoted statistically significant increases in rod cell volumes (*Figure 3—figure supplement 2B*). These data suggest some lead compounds were able to maintain rod outer segment morphology, thus potentially maintaining rod cell function.

## Mouse model validation I: primary photoreceptor cells treated with stressor compounds

To identify new therapeutics for RP patients, we reasoned compounds producing neuroprotective effects across fish and mammalian RP models would likely target conserved mechanisms, and thus be more likely to translate successfully in the clinic. We therefore tested the efficacy of lead compounds in a series of mouse models of retinal photoreceptor degeneration.

First, we tested compounds for the capacity to protect mouse primary photoreceptor cells from stressor-induced cell death in culture. Photoreceptors were isolated from postnatal day four (P4) QRX mice, a transgenic line in which GFP expression is restricted to photoreceptors (*Wang et al., 2004*), and grown as previously described (*Fuller et al., 2014*). To induce photoreceptor cell death, thapsigargin (0.25 µM) or tunicamycin (0.6 µg/mL) were added. These 'stressor' compounds deplete endoplasmic reticulum (ER) calcium levels (*Thastrup et al., 1990*) and inhibit protein glycosylation (*Fliesler et al., 1984*), respectively (*Lai et al., 2007b*). In turn, they elicit an unfolded protein response (UPR) (*Wang et al., 2015b*) and related ER stress (*Oslowski and Urano, 2011*; *Zhang et al., 2014*), both implicated in the etiology of RP (*Griciuc et al., 2011*; *Rana et al., 2014*). To test for survival effects, all eleven lead compounds were screened at seven concentrations across a threefold dilution series (from 30 µM to 40 nM). After 48 hr in stressor ± lead compounds, cells were imaged using an automated high-content screening system. Photoreceptor survival was assessed by automated quantification of GFP-expressing cells (*Figure 4A*). Nine of 11 lead compounds, proved protective in the thapsigargin-induced cell death assay (*Figure 4B*). MIC, CLO and SUL also protected photoreceptors from tunicamycin-induced cell death (*Figure 4—figure supplement 1*). Thus, nine of 11 lead compounds promoted the survival of mouse primary photoreceptor cells in at least one stressor-induced cell death assay, and three were neuroprotective in both assays.

## Mouse validation II: rd1 mutant retinal explants

We next tested lead compounds for survival effects in *rd1* mouse retinal explant cultures. The *rd1* mouse model of RP exhibits early onset rod cell degeneration caused by a mutation in the *Pde6b* gene (*Danciger et al., 1990*; *Pittler and Baehr, 1991*; *Sidman and Green, 1965*), which is an ortholog of the human RP-associated *PDE6B* (*Khramtsov et al., 1993*; *McLaughlin et al., 1993*). In *rd1* mice, photoreceptor degeneration begins around P10 and progresses rapidly. By P21, only a single row of photoreceptor cells remain in the outer nuclear layer (ONL; *Tansley, 1951*) making it an excellent system for screening potential neuroprotectants (*Beeson et al., 2016*). Here, retinal explants from P10 *rd1* mice were isolated and cultured ex vivo (*Bandyopadhyay and Rohrer, 2010*). Eight lead compounds, all but CAL, SUL, and Zinc pyrithione (ZPT), were tested for survival effects

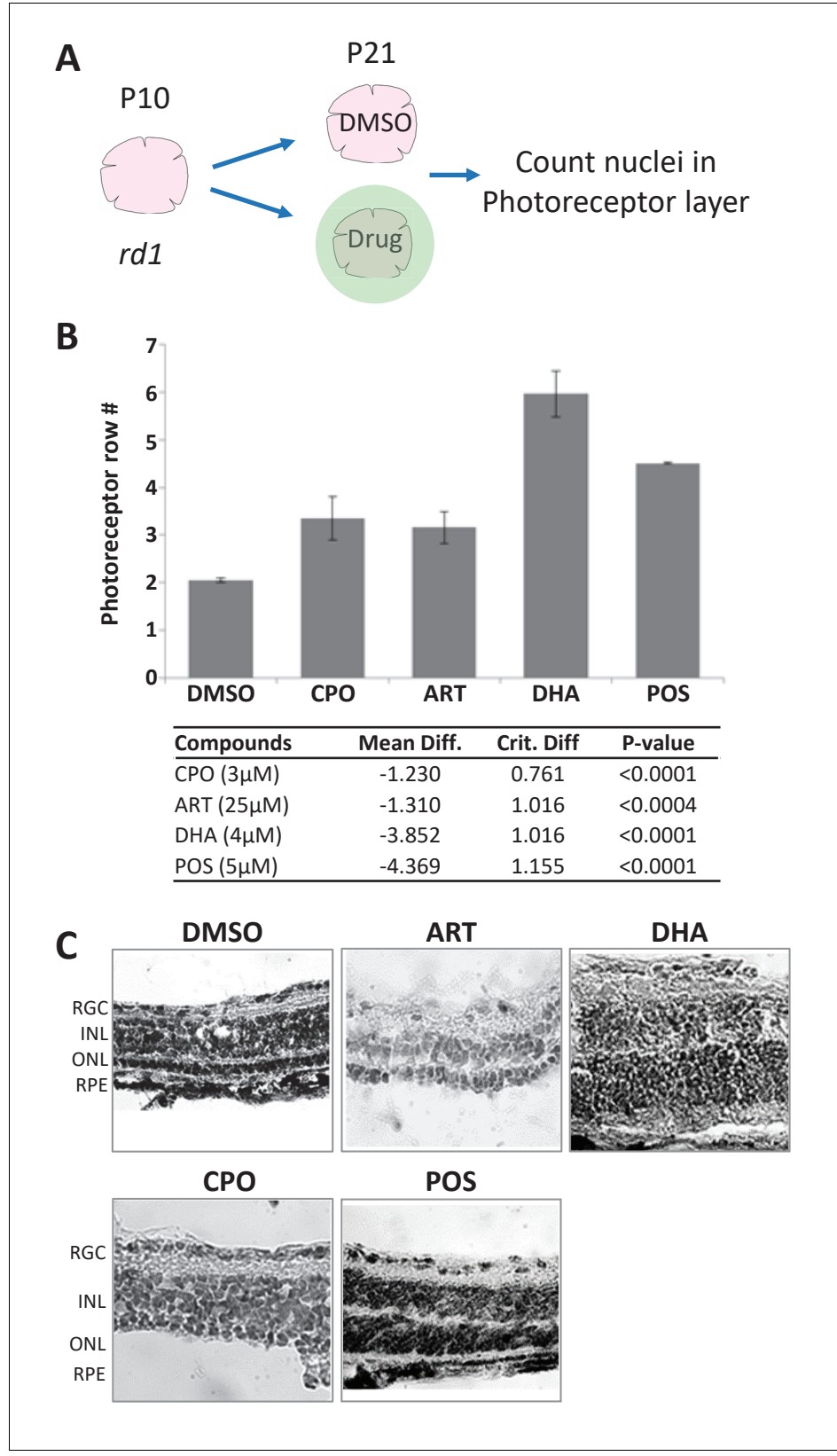

**Figure 5.** Lead effects on mouse *rd1* retinal explant cultures. (**A**) Diagram of *rd1* explant photoreceptor survival assay protocol. Retinal explants were isolated at postnatal day 10 (**P10**) and cultured for eleven days in lead compounds at three different concentrations or DMSO. The number of surviving photoreceptor rows was then
*Figure 5 continued on next page*

*Figure 5 continued*

quantified. (**B**) Maximally effective dosage of ciclopirox olamine (CPO), artemisinin (ART), and dihydroartemisinin (DHA). Repeated measures ANOVA followed by Bonferroni/Dunn posthoc correction was used to assess statistical significance. (**C**) Represented images of retinal explants under conditions listed. INL, inner nuclear layer; ONL, outer nuclear layer; RGC, retinal ganglion cell layer; RPE, retinal pigment epithelium.
The online version of this article includes the following source data for figure 5:

**Source data 1.** Compound tests - rd1 mouse retinal explants.

at three concentrations across a fivefold dilution series centered on the concentration most effective in fish RP models. After 11 days in culture, explants were fixed, stained and the number of photoreceptor rows counted (*Figure 5A*). Neuroprotective effects were defined as a concentration-dependent increase in the number of photoreceptor rows remaining in the ONL relative to untreated controls (p≤0.05). An average of 1.2 ±0.19 rows of photoreceptors remained in the ONL of control explants. Three lead drugs, CPO, DHA, and ART, increased the number of surviving photoreceptor layers (*Figure 5B*). Representative images of control (DMSO and POS) and lead-treated explants demonstrate photoreceptor layer protection (*Figure 5C*). CPO, DHA, and ART thus promoted rod cell survival in the fish RP model and two mouse cell culture RP models. However, high concentration CPO treatments (15 µM) led to disruption of retinal histology due to induction of proliferation in the inner nuclear layer (INL) and ONL. Therefore, as DHA is the active metabolite of ART-related compounds, we proceeded with testing DHA in an in vivo mouse RP model.

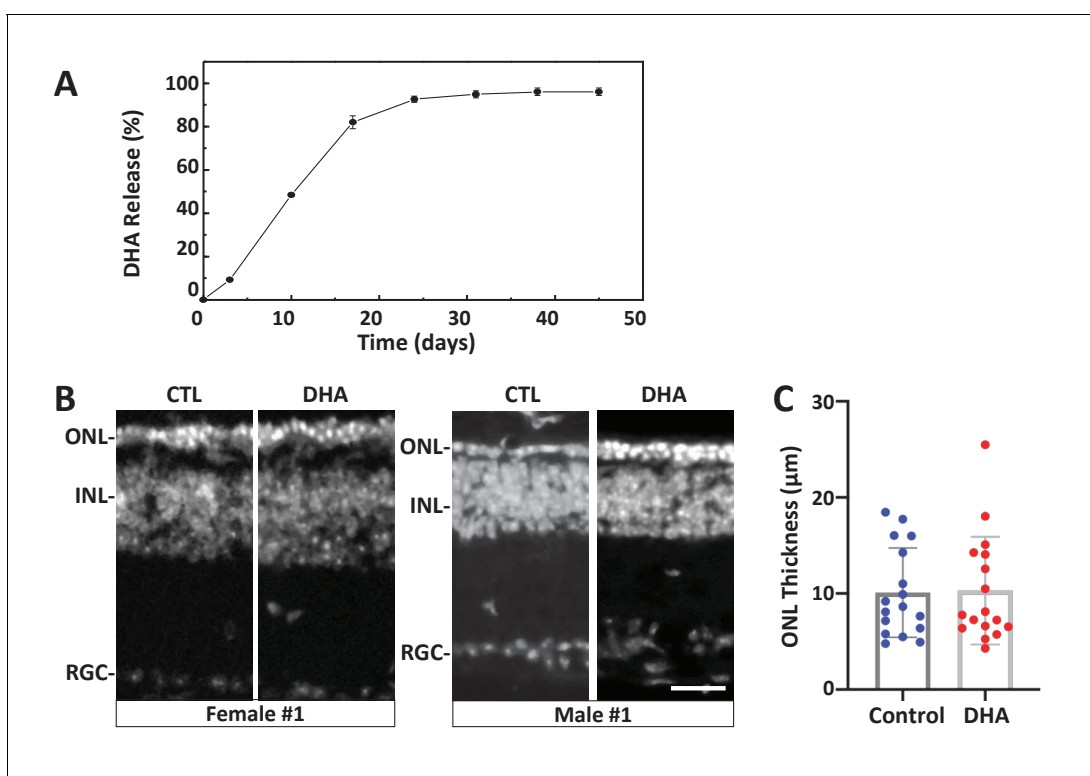

**Figure 6.** Test of long-release DHA formulation in the *rd10* mouse model of RP. (**A**) In vitro release kinetics of a long-release DHA formulation (DHA encapsulated in PLGA polymer microparticles, PLGA-DHA) in phosphate buffered saline containing 0.1% DMSO (pH 7.4) at 37°C (*Figure 6—source data 1*). (**B**) Representative images of DAPI-stained retinal sections from one female and one male *rd10* retina injected with vehicle control (CTL) or PLGA-DHA (DHA; scale bar = 25 µm). (**C**) Quantification of ONL thickness of control and PLGA-DHA-treated retinas, no statistically significant differences were observed (p=0.5898, two-way ANOVA). The total sample size was 17 mice across three experimental replicates. Abbreviations: CTL, control; DHA, Dihydroartemisinin; ONL: outer nuclear layer; INL: inner nuclear layer; GCL: ganglion cell layer.
The online version of this article includes the following source data for figure 6:

**Source data 1.** PLGA-DHA in vitro release kinetics.

**Table 2.** Summary of PubChem HTS and uHTS target-based screens of lead compound bioactivity.

Results of 13 target-based HTS screens, with '+' indicating inhibitory activity of lead compounds. Lead compound abbreviations: WAR, Warfarin; CLO, Cloxyquin; CPO, Ciclopirox olamine; MIC, Miconazole; ZPT, Zinc pyrithione; DHA, Dihydroartemisinin; CHL, Chloroxine; CAL, Calcimycin; SUL, Sulindac; ART, Artemisinin; COR, Cortexolone; POS, positive control. Other abbreviations: CI, confidence interval; Mtz, Metronidazole; PTU, N-phenylthiourea. Target abbreviations: Tdp1, Tyrosyl-DNA phosphodiesterase 1; Rorc, RAR-related orphan receptor gamma; AR, androgen receptor signaling pathway; TR, thyroid receptor signaling pathway; VDR, vitamin D receptor; ER, estrogen receptor alpha; AhR, aryl hydrocarbon receptor; GR, glucocorticoid receptor; Dopa, dopamine related; HIF1, Hypoxia-inducible factor 1-alpha; SHH, Sonic hedgehog; COX, cyclooxygenase.

| Drug\path | Tdp1 | Rorc | AR | TR | VDR | ER | AhR | GR | P53 | Dopa | HIF1 | SHH | COX |
|---|---|---|---|---|---|---|---|---|---|---|---|---|---|
| WAR | | | | | | | | | + | | | | |
| MIC | + | + | | + | + | | + | | | | | | |
| CLO | + | + | + | + | + | + | | | | + | | | |
| CPO | + | + | + | + | + | + | | + | + | + | | | |
| ZPT | + | + | + | + | + | + | + | + | + | | + | | |
| DHA | + | | + | + | | + | | | | | | | |
| CHL | + | + | + | + | + | + | + | + | + | | | | |
| CAL | | + | | | | | + | | | | | + | |
| SUL | + | | | | | | + | | | | | | + |
| ART | + | | | | | | | | | + | | | |
| COR | | | + | | | | | | | + | | | |

**Table 3.** Chemical inhibitors of lead implicated targets and cell death pathways.

List of eight PARP inhibitors, one necroptosis inhibitor, three apoptosis inhibitors, and three TDP1 inhibitors tested for neuroprotective effects in *rho:YFP-NTR* larvae (see *Figure 4* and *Figure 4—source data 1*). Target(s) and relative inhibitory activity against related targets in each cell death pathway are as reported by manufacturer, references provided by manufacturer, or references herein. Concentrations producing maximal survival effects and the percent increase in YFP levels are listed for each compound. Abbreviations: Parp (Poly (ADP)-ribose Polymerase), Ripk1 (Receptor-interacting serine/threonine-protein kinase 1), Tdp1 (Tyrosyl-DNA phosphodiesterase 1).

| Cell death Pathway | Compound name (abbrv.) | Target(s) and Relative Activity (+) | Conc. [μM] | Effect (%) |
|---|---|---|---|---|
| Parp-dependent (Parthantos or cGMP-dependent) | AG-14361 (AG) | Parp1+++ | 64 | 9 |
| | NMS-P118 (NMS) | Parp1++ | 8 | 16 |
| | Talazoparib (BMN) | Parp1 ++++ | 0.5 | 14 |
| | Veliparib (ABT) | Parp1+++, 2+++ | 64 | 9 |
| | Olaparib (Ola) | Parp1+++, 2++++ | 32 | 9 |
| | E7449 | Parp1++++, 2++++ | 1 | 20 |
| | Niraparib (MK) | Parp1+++, 2+++ | 32 | 11 |
| | Rucaparib (RUC) | pan-Parp++++ | 32 | 17 |
| Necroptosis | Necrostatin-1 (NEC) | Rip1k++ | 64 | 17 |
| Apoptosis | Ac-DEVD-CHO (AC) | Caspase 1+++, 2+, 3++++, 4++, 5++, 6+++, 7+++, 8++++, 9++, 10+++ | 0.5 | 3 |
| | Caspase3/7 inhibitor I (CASI) | Caspase 3++, 7++, 9+ | 0.5 | 1 |
| | Caspase three inhibitor VII (CASVII) | Caspase 3+++ | 1 | 2 |
| Tdp1 (DNA repair) | Paromomycin (PM) | Tdp1+ | 20 | 10 |
| | Thiostrepton (ThS) | Tdp1+ | 100 | 10 |
| | Methyl-3,4-dephostatin (MD) | Tdp1++ | 0.16 | 3 |

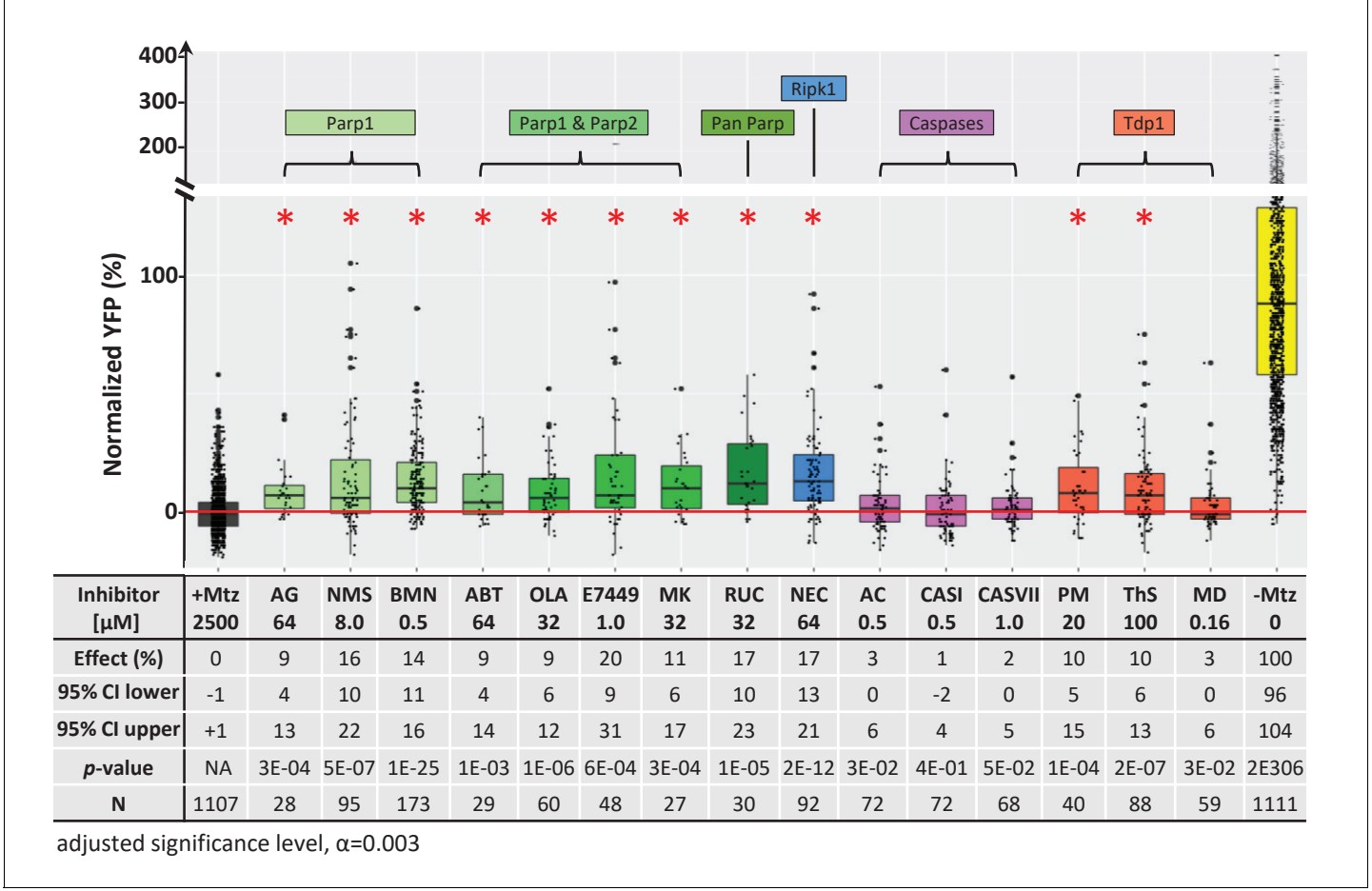

**Figure 7.** Chemical inhibitor analysis of NTR/Mtz-mediated rod cell death in zebrafish. Box plots of rod cell survival effects of eight PARP inhibitors (green), a necroptosis inhibitor (blue), four apoptosis inhibitors (magenta), and three Tdp1 inhibitors (orange) in Mtz-treated *rho:YFP-NTR* zebrafish larvae (targets are listed above for each category). Assays performed as per confirmation tests (see *Figure 2A*) with all inhibitors tested across 10 concentrations using a twofold dilution series (see Methods for highest concentration tested per compound). Conditions resulting in a statistically significant increase in survival effects relative to +Mtz controls are marked with an asterisk. Survival effects (normalized YFP %), 95% confidence intervals, p-values, and sample sizes (N) for each condition are shown in the table below. Student's *t*-test was used to calculate p-values for each condition relative non-ablated controls (0 mM Mtz). Bonferroni correction for multiple comparisons resulted in an adjusted significance level of 0.003 ($\alpha$=0.003). A minimum of two experimental repeats was performed for each condition (*Figure 7—source data 1*). No statistical differences in larval survival were observed for tested compounds relative to their respective +Mtz controls, except for PM (67%; Fisher's exact test, p<0.05). Target abbreviations: Parp, Poly (ADP)-ribose polymerase; Ripk1, Receptor-interacting serine/threonine-protein kinase (1); Tdp1, Tyrosyl-DNA phosphodiesterase 1; Mtz, metronidazole. Inhibitor abbreviations: AG, AG-14361; NMS, NMS-P118; BMN, talazoparib; ABT, veliparib; OLA, olaparib; MK, niraparib; RUC, rucaparib; NEC, necrostatin-1; AC, Ac-DEVD-CHO; CASI, caspase3/7 inhibitor I; CASVII, caspase three inhibitor VII; PM, paromomycin; ThS, thiostrepton; MD, methyl-3,4-dephostatin. Other abbreviations: CI, confidence interval; Mtz, Metronidazole.
The online version of this article includes the following source data and figure supplement(s) for figure 7:

**Source data 1.** Cell death inhibitor assay.
**Figure supplement 1.** Additive survival effects of paired PARP and necroptosis inhibitors.
**Figure supplement 1—source data 1.** Paired cell death inhibitor assay.

## Mouse validation III: rd10 mutant model of RP

To test the potential of DHA as a 'pan-disease' therapeutic, that is, whether it was effective across multiple genetic RP models, it was tested for photoreceptor survival effects in the *rd10* mouse model of RP (*Pde6b^rd10^*). The *rd10* line was selected for in vivo experiments due to its slower rate of photoreceptor degeneration relative to other *rd* mutants, thus allowing a prolonged window for pharmacological intervention. DHA was selected based on its superior performance in *rd1* retinal explant assays (*Figure 5*) and because it is the active metabolite of a second lead drug candidate,

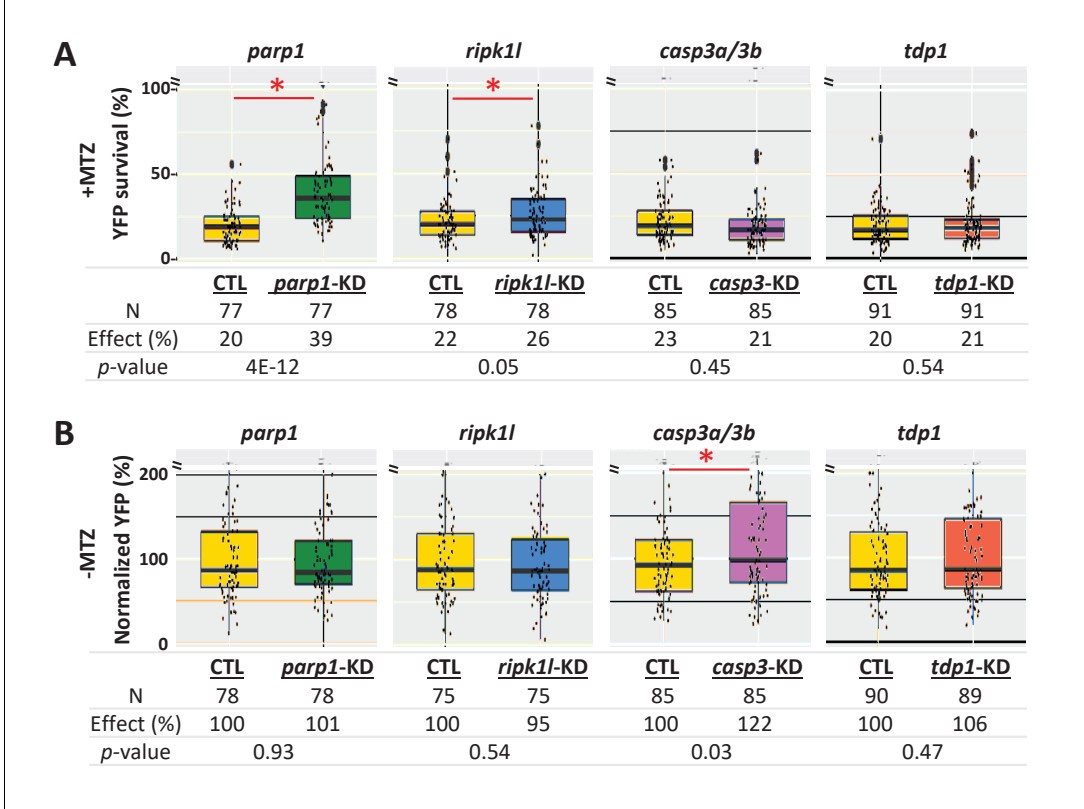

**Figure 8.** Genetic knockdown analysis of NTR-mediated rod cell death pathways in zebrafish. (**A**) Box plots of rod cell survival effects of CRISPR/Cas9-based knockdown of key cell death pathway genes: *parp1* (parthanatos), *ripk1l* (necroptosis), *casp3a* and *casp3b* (apoptosis), as well as *tdp1* (as a DNA repair pathway control) in Mtz-treated *rho:YFP-NTR* zebrafish larvae. Cas9 protein and four gRNAs (for *parp1*, *ripk1l* and *tdp1*) or eight gRNAs (for *casp3a* with *casp3b*) were co-injected at the one-cell stage. Larvae were raised to 5 dpf and treated with Mtz (2.5 mM) to induce rod cell death. Rod cell survival was quantified at 6 dpf by fluorescence microplate reader assay. Conditions resulting in a statistically significant increase in survival effects relative to non-injected controls are marked with an asterisk. Sample sizes (**N**), survival effects (YFP % relative to non-ablated control) and p-values are provided in the table below. A minimum of three experimental repeats was performed and data pooled across replicates (*Figure 8—source data 1*). (**B**) Box plots of effects on rod cell development in non-ablated *rho:YFP-NTR* zebrafish larvae (-Mtz) following knockdown of the same genes as in (**A**). Rod cell numbers were estimated at 6 dpf by microplate reader-based quantification of YFP. Only *casp3a/3b* knockdown resulted in a statistically significant increase (22%) in rod cell numbers during development. Sample sizes (**N**), normalized survival effects (% relative to non-injected control) and p-values are provided in the table below. A minimum of three experimental repeats was performed and data pooled across replicates. Abbreviations: *parp1*, poly (ADP-ribose) polymerase 1; *ripk1l*, receptor (TNFRSF)-interacting serine-threonine kinase 1, like; *casp3a*: caspase 3, apoptosis-related cysteine peptidase a; *casp3b*: caspase 3, apoptosis-related cysteine peptidase b; *tdp1*, tyrosyl-DNA phosphodiesterase 1; CTL, control; KD, knockdown; PAR, Poly (ADP-ribose).

The online version of this article includes the following source data and figure supplement(s) for figure 8:

**Source data 1.** Quantification of PAR accumulation.

**Figure supplement 1.** Genetic and biochemical analysis of NTR-mediated rod cell death in zebrafish.

**Figure supplement 1—source data 1.** qPCR confirmation of gene knockdown.

ART, recently shown to protect photoreceptors from light damage (*Lu and Xie, 2019*). To create a long-release formulation, DHA was encapsulated in PLGA polymers (Poly(D,L-lactide-co-glycolide)). Release kinetics assays in PBS/0.1% DMSO in vitro suggested DHA would reach maximal concentrations after 30 days, and remain stable for at least 20 days thereafter (*Figure 6A*). We noted that release was minimal in the absence of DMSO, but were reluctant to include DMSO for injections due to the potential for deleterious effects on the retina (*Tsai et al., 2009*). PLGA-DHA was injected into the vitreous of one eye of *rd10* mice at P14, with the contralateral eye serving as a vehicle injection control. At P32, 18 days after injection, and when DHA levels were predicted to reach 80% of maximal concentration based on in vitro releases kinetics, *rd10* mouse eyes were processed for immunohistochemistry (*Figure 6B*) and ONL thickness quantified (*Figure 6C*). Despite initial promising

results in pilot assays, no reproducible neuroprotective effects were observed. It is possible that in the absence of DMSO release kinetics may have been limiting in vivo. We plan to address this possibility in future studies.

## Molecular mechanism of action

Previously, we used 'on label' information to explore MOA of hit compounds identified during an in vivo phenotypic screen of the JHDL (*Wang et al., 2015b*). Recently, we have become interested in additional advantages afforded by HTS-based MOA data and whole-organism phenotypic screening, such as polypharmacology (*Dar et al., 2012*; *Rennekamp and Peterson, 2015*; *Rihel et al., 2010*). Accordingly, we applied a target-agnostic MOA analysis process by evaluating lead compound performance in HTS and ultra-HTS studies archived on PubChem (https://pubchem.ncbi.nlm.nih.gov/). Many lead compounds exhibited shared target/pathway activities, suggesting in-common MOA (*Table 2*). Among shared targets, Tyrosyl-DNA Phosphodiesterase 1 (TDP1), a DNA repair enzyme, was the most popular (eight leads). To test whether TDP1 inhibition promoted rod photoreceptor survival, we tested three TDP1 inhibitors in *rod:YFP-NTR* fish: paromomycin (PM), thiostrepton (ThS), and methyl-3,4-dephostatin (MD) (*Huang et al., 2011*; *Liao et al., 2006*). Both PM and ThS showed neuroprotective effects (*Table 3*, *Figure 7*).

This result was surprising given that NTR reduction of Mtz is thought to cause DNA damage-induced cell death (*Curado et al., 2007*). Thus, inhibition of a DNA repair enzyme would be expected to enhance, not inhibit, NTR/Mtz-mediated cell death. However, an alternative means of disrupting TDP1 activity is by inhibiting Poly (ADP-ribose) Polymerases (PARPs). Indeed, in a comprehensive 400,000 compound qHTS assay designed to identify TDP1 inhibitors, all five hits turned out to be PARP inhibitors (*Murai et al., 2014*). PARPs also mediate DNA repair but, interestingly, hyperactivation of PARP1 leads to a specific form of DNA damage-induced cell death, termed parthanatos (*Fatokun et al., 2014*; *Wang et al., 2016*). PARP is also involved in a cGMP-dependent cell death pathway implicated in RP (*Power et al., 2020*).

We therefore assayed PARP inhibitors for the capacity to promote rod cell survival in *rod:YFP-NTR* fish. All eight PARP inhibitors tested had neuroprotective activity, ranging from 9% to 20% (*Figure 7*). To account for other cell death mechanisms implicated in neurodegeneration, we tested an inhibitor of necroptosis (necrostatin-1; NEC), and three inhibitors of apoptosis (Ac-DEVD-CHO, CASI, and CASVII, *Table 3*). Surprisingly, NEC promoted rod cell survival, while the apoptosis inhibitors did not (*Figure 7*, *Table 3*). Interestingly, paired testing of PARP and necroptosis inhibitors resulted in additive effects (*Figure 7—figure supplement 1*, *Supplementary file 1b*), suggesting either differential responses to NTR/Mtz-induced DNA damage among rod cells, or that inhibiting both pathways keeps cells from activating the other mechanism of cell death when only one pathway is blocked.

To further dissect cell death pathways mediating NTR/Mtz-induced rod cell ablation, we used a 'redundant' CRISPR/Cas9 gene targeting approach (i.e. four guide RNA targeting), which enables phenotyping in injected zebrafish embryos/larvae (*Wu et al., 2018*). Key factors for apoptosis (*casp3a* and *casp3b*), PARP-dependent cell death pathways parthanatos and/or cGMP-dependent cell death (*parp1*), and necroptosis (*ripk1l*) were targeted. In addition, *tdp1* was targeted as a DNA repair control. For each targeted gene, four gRNAs and Cas9 protein were co-injected into one-cell stage embryos. Injected and uninjected (control) larvae were treated ±2.5 mM Mtz at five dpf and rod-YFP levels quantified at six dpf by ARQiv. For Mtz-ablated larvae, knockdown of *parp1* markedly increased rod cell survival (39% compared to 20% in the uninjected control, $p=4E-12$), and *ripk1l* knockdown modestly protected rods (26% compared to 22% in the uninjected control, $p=0.05$). However, no improvement in survival was observed for either *casp3a/b* or *tdp1* knockdown (*Figure 8A*). In no Mtz controls, knockdown of *parp1*, *ripk1l* or *tdp1* had no effect on rod cell numbers. Conversely, *casp3a/3b* double knockdown increased rod cell numbers (*Figure 8B*) suggesting inhibition of developmental apoptosis and confirming efficacy of *casp3a/3b* knockdown. Confocal imaging confirmed *parp1* knockdown effects on rod cell survival and *casp3a/3b* knockdown effects on rod cell development (*Figure 8—figure supplement 1A*). Reduced gene expression was verified by qPCR (*Figure 8—figure supplement 1B*, *Supplementary file 3*). These results are consistent with the NTR/Mtz system eliciting DNA damage-associated cell death mediated by *parp1*, that is, parthanatos, in rod cells.

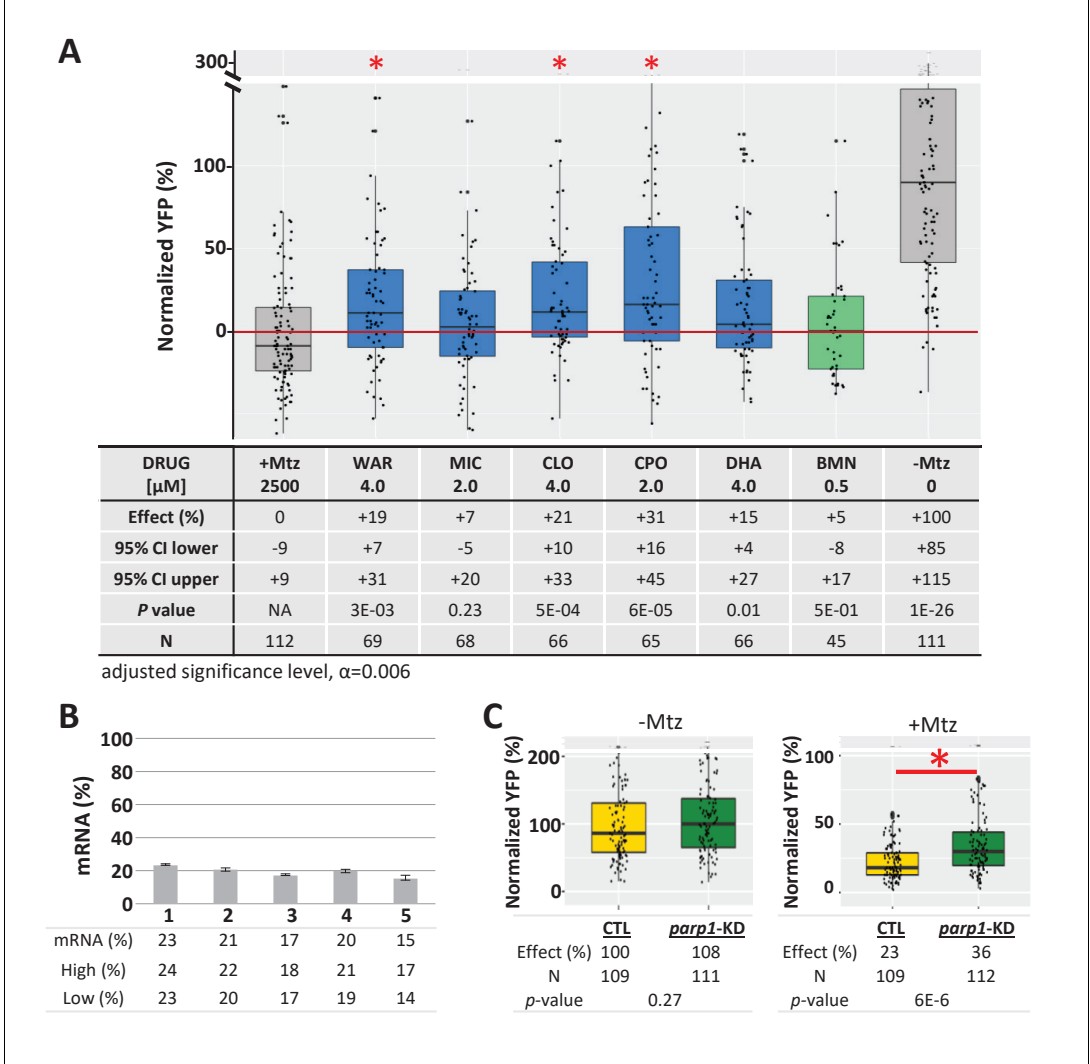

**Figure 9.** Test of lead compound survival effects in *parp1* knockdown zebrafish larvae. (A) Box plots of rod cell survival effects of five lead compounds and a PARP inhibitor control (BMN) in Mtz-treated *rho:YFP-NTR* zebrafish larvae in which *parp1* expression was knocked down, assays performed as per confirmation tests (see *Figure 2A*). Larvae subjected to CRISPR-based knockdown of *parp1* were raised to 5 dpf, exposed to lead compounds at the indicated concentrations, and treated with 2.5 mM Mtz to induce rod cell death. Rod cell survival was quantified at six dpf by fluorescence microplate reader assay. Conditions that promoted a statistically significant increase in rod cell survival are marked with an asterisk. Survival effects (normalized YFP %), 95% confidence intervals, p-values, and sample sizes (N) for each condition are shown in the table below. Student's *t*-test was used to calculate p-values for each condition relative to ablated controls (+Mtz). Bonferroni correction for multiple comparisons resulted in an adjusted significance level of 0.006 (α=0.006). A minimum of three experimental repeats were performed for each condition and data polled across replicates (*Figure 9—source data 1*). (B) Quantification of CRISPR-based *parp1* knockdown, 16–24 larvae per condition (seven dpf) were pooled as one sample for RNA extraction. qPCR results confirmed suppression of parp1 expression (ranging from 14% to 23% of uninjected controls). (C) Box plots of *parp1* knockdown effects on rod cell neogenesis and rod cell survival, that is, -Mtz and +Mtz controls, respectively, for these experiments. Sample sizes (N), survival effects (normalized YFP %) and p-values are shown in the table below. Lead compound abbreviations: WAR, Warfarin; CLO, Cloxyquin; CPO, Ciclopirox olamine; MIC, Miconazole; DHA, Dihydroartemisinin; Other abbreviations: BMN, talazoparib (*parp1*-dependent control);, CI, confidence interval; Mtz, Metronidazole; *parp, poly (ADP-ribose) polymerase 1*.

The online version of this article includes the following source data for figure 9:

**Source data 1.** Compound effects following gene knockdown.

To further test for activation of PARP signaling during NTR/Mtz-induced rod cell death, accumulation of polymerized poly (ADP-ribose) (PAR), a downstream effector of *PARP1* activation (*Kam et al., 2018*), was analyzed by western blot. five dpf *rho:YFP-NTR* larvae were treated ±2.5 mM Mtz and protein samples collected from ~30 fish at 3, 6, 12, 24, and 48 hr post-Mtz treatment initiation.

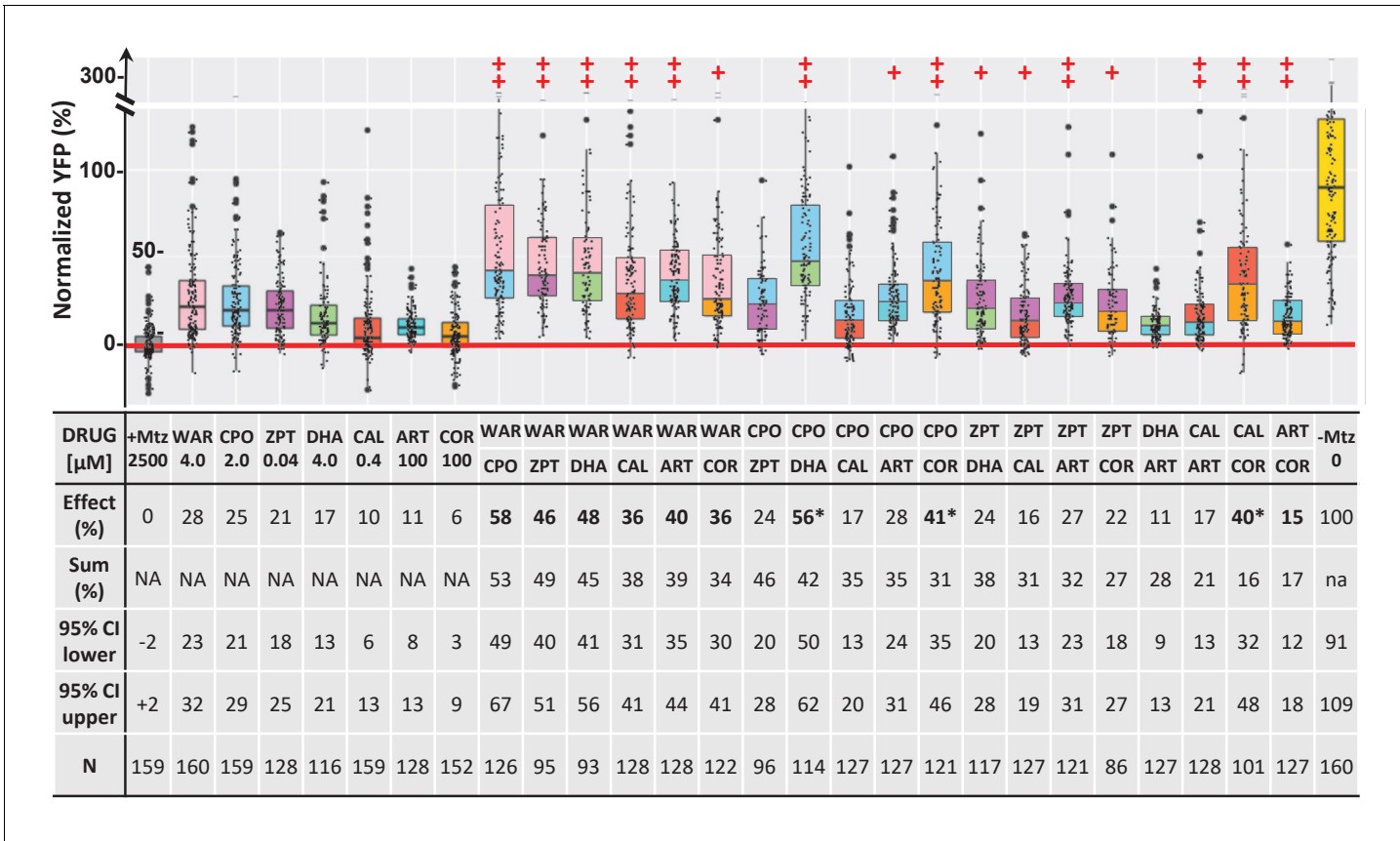

**Figure 10.** Additive survival effects of paired lead compounds in zebrafish RP model. Box plots of rod cell survival effects of seven lead compounds tested alone and in pairs in *rho:YFP-NTR* zebrafish larvae (assays performed as per confirmation tests, see *Figure 2A*). To test for enhanced survival, all compounds were tested at maximal effective dosages. Eleven pairs promoted a statistically significant increase in survival relative to both individual compound controls (++), while five pairs performed better than a single compound control (+). Survival effects (normalized YFP %), 95% confidence intervals, p-values, and sample sizes (N) for each condition are shown in the table below. For comparison, the survival effect of each pair (Effect %) and the summed effect of the respective individual compound controls (Sum %) are provided. Additive effects were defined as any pair that promoted survival at equal to or greater than the sum of the corresponding individual compound effects ±10%. By this criterion, 10 pairs produced additive effects (bolded values). Three pairs produced effects greater than 25% above the summed effect and were considered potentially synergistic (asterisks). A minimum of three experimental repeats were performed for each condition and data pooled across replicates (*Figure 10—source data 1*). No statistical differences in larval survival were observed for lead compounds or compound pairs relative to their respective +Mtz controls, except for DHA (73%), WAR+DHA (73%), CIC+DHA (89%), ZPT+DHA (91%), ZPT+COR (90%) and CAL+COR (79%; Fisher's exact test, p<0.05). Two pairs, DHA+CAL and DHA+COR resulted in total lethality and are not plotted. See *Supplementary file 1c* for survival effects (normalized YFP %), 95% confidence intervals and p-values relative to ablated controls (+Mtz) and to respective individual compound controls, and sample sizes (N) for all conditions. Lead compound abbreviations: WAR, Warfarin; CLO, Cloxyquin; CPO, Ciclopirox olamine; MIC, Miconazole; ZPT, Zinc pyrithione; DHA, Dihydroartemisinin; CHL, Chloroxine; CAL, Calcimycin; SUL, Sulindac; ART, Artemisinin; COR, Cortexolone; POS, positive control. Other abbreviations: CI, confidence interval; Mtz, Metronidazole.

The online version of this article includes the following source data for figure 10:

**Source data 1.** Paired compound tests.

Western blots showed clear accumulation of PAR in Mtz-treated fish at 24 and 48 hr post-Mtz (*Figure 8—figure supplement 1C*) which was verified quantitatively (*Figure 8—figure supplement 1D*). Collectively, results of cell death pathway analyses are concordant across chemical inhibition, genetic knockdown, and biochemical assays, strongly suggesting NTR/Mtz-induced cell ablation is mediated by PARP-dependent cell death (e.g. parthanatos, cGMP-dependent cell death), a target/pathway implicated across multiple mammalian models of RP (*Arango-Gonzalez et al., 2014*).

To determine if lead compound effects were mediated by inhibition of PARP signaling, *parp1* expression was knocked down prior to testing for effects on rod cell survival in *rho:YFP-NTR* larvae. Four leads predicted to be PARP inhibitors (MIC, CLO, CPO, DHA), one predicted to be a non-PARP inhibitor (WAR), and a known PARP inhibitor control (BMN) were tested. Two lead compounds, MIC and DHA and BMN (the PARP inhibitor control) produced no statistically significant enhanced neuroprotective effects over *parp1* knockdown. Conversely, WAR, CLO and CPO increased rod cell survival in *parp1* knockdown fish (*Figure 9A*). qPCR confirmed mRNA knockdown (*Figure 9B*) and controls showed no effects of *parp1* knockdown on rod cell development (*Figure 9C*, -Mtz) and enhanced rod cell survival upon Mtz treatment (*Figure 9C*, +Mtz). These data confirm that PARP inhibition mediates the neuroprotective effects of some lead compounds (e.g. MIC and DHA) while others are PARP-independent. The latter result suggests lead compounds may act via complementary neuroprotective mechanisms. Combined, these findings are consistent with rod cell loss in our fish RP model occurring, at least partially, by PARP-dependent cell death.

## Combinatorial assay

PubChem data also suggested potential complementary MOA, that is, multiple independent targets across lead compounds (*Table 2*). We therefore hypothesized that combining lead compounds may produce enhanced, additive, or even synergistic effects. To investigate this, combinatorial lead drug treatments were tested in primary mouse photoreceptor cell cultures. As above, photoreceptors were isolated from QRX mouse retinas and used in stressor-treated primary cell culture assays. All 11 lead compounds were tested at a selected concentration individually and in pairs and the survival of GFP-labeled photoreceptors analyzed by high-content imaging and automated cell counting. The results showed no wholly additive effects, that is, survival percentages equaling or excelling the summed effect of individual compound assays. However, enhanced effects – i.e., better median survival than for either compound alone – were observed for 30 of 55 pairs in thapsigargin-treated cultures and for 20 of 55 pairs in tunicamycin-treated cultures (*Figure 4—figure supplement 2*). Finally, seven lead compounds were tested alone and in pairs in *rho:YFP-NTR* zebrafish using optimal effective concentrations. Pairs producing survival effects equal to or greater than the sum of their individual values (±10%) were considered additive. By this criterion, 10 of 19 viable pairs (two pairs proved lethal) exhibited additive effects (*Figure 10*, bolded; *Supplementary file 1c*) and three pairs produced supra-additive effects (i.e. ≥25% greater than summed effects) suggesting possible synergy (*Figure 10*, asterisks). The maximal average rod cell survival effect was 58% (WAR + CPO). Several compounds showed broadly additive effects, for example, WAR was additive with all six drugs and COR with four of five pairs (one pair proving lethal). Additive effects suggest multiple signaling pathways are involved in NTR/Mtz-induced photoreceptor degeneration, thus that combinatorial drug regimens may be required to achieve maximal therapeutic benefits for RP patients.

## Discussion

Identifying effective neuroprotective therapies for RP and other IRDs stands as a critical unmet need for the field (*Duncan et al., 2018*; *Wubben et al., 2019*). Although, neurotrophic factors, anti-apoptotic agents, nutritional supplements, and antioxidants have shown neuroprotective effects in animal models of RP (*Dias et al., 2018*). Unfortunately, these reagents have produced, at best, only limited benefits for patients to date and, for some, mild improvements are offset by adverse side effects associated with long-term use (*Dias et al., 2018*). For example, ciliary neurotrophic factor (CNTF) was shown to be effective in protecting photoreceptors in mouse (*Cayouette et al., 1998*), dog (*Tao et al., 2002*), and chicken (*Fuhrmann et al., 2003*) models of retinal degeneration. However, CNTF failed to improve either visual acuity or field sensitivity in short- and long-term RP clinical trials (*Birch et al., 2016*; *Ciliary Neurotrophic Factor Retinitis Pigmentosa Study Groups et al., 2013*; *Argus II Study Group et al., 2015*). Clinical trials of Vitamin A in combination with Vitamin E (*Berson et al., 1993*), docosahexaenoic acid (*Berson et al., 2004*), lutein (*Berson et al., 2010*), or valproic acid (*Birch et al., 2018*) were reported to produce some benefits for RP patients, but only in subpopulations, and some of these studies have been controversial (*Massof and Finkelstein, 1993*).

Our strategy for addressing this challenge was to: (1) scale up the number of compounds tested directly in complex living disease models and (2) perform a cross-species screening cascade that

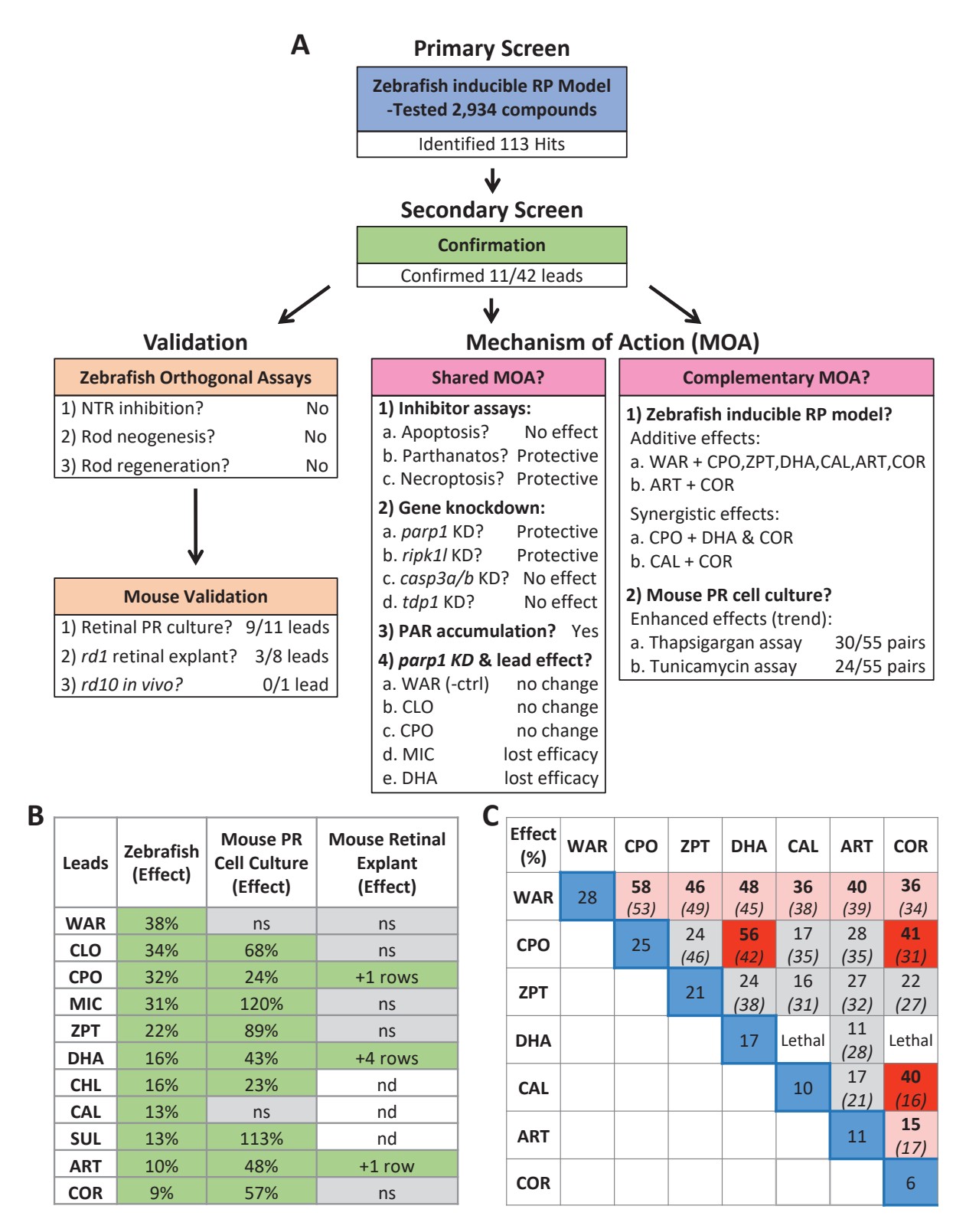

**Figure 11.** Summary. (**A**) Flow chart of cross-species phenotypic drug discovery process showing results of the primary drug screen, secondary confirmation, orthogonal assays, mouse model tests, shared MOA and complementary MOA assays (i.e. paired drug tests). (**B**) Summary of survival effects of lead drug candidates across model systems. (**C**) Summary of survival effects of paired lead drug tests in the zebrafish-inducible RP model showing effects of single drug controls (blue boxes) and paired drug tests that resulted in no additive effects (gray boxes), additive effects (orange

*Figure 11 continued on next page*

*Figure 11 continued*

boxes) and synergistic effects (red boxes); mathematically summed effect of each pair of lead compounds (italics) are shown for comparison. Abbreviations: as in prior figures.

starts with small animal models amenable to HTS and proceeds to mammalian models. We hypothesize that compounds producing beneficial outcomes across evolutionarily diverse species will target conserved MOA and thus stand a higher chance of successful translation. As a generalized strategy, large-scale drug discovery screens using small animal models are showing increasing promise across multiple disease paradigms (*Cagan et al., 2019*; *Cully, 2019*; *Kitcher et al., 2019*; *MacRae and Peterson, 2015*).

Here, using a large-scale in vivo drug screening platform (*Figure 1*), we tested 2934 largely human-approved compounds for neuroprotective effects across six concentrations in >350,000 larval zebrafish models of RP. The primary screen implicated 113 compounds as neuroprotectants (*Supplementary file 2c*). Confirmatory repeats and a series of four orthogonal assays validated 11 of 42 prioritized hit compounds in protecting zebrafish rod photoreceptors from cell death (*Figures 2* and *3*, *Figure 2—figure supplements 1–3*, *Figure 3—figure supplement 1* and *Table 1*). Importantly, investigations of lead compound MOA, led to the discovery that NTR/Mtz-mediated rod cell death appears to proceed through alternative cell death pathways (*Figure 8*) recently linked to photoreceptor degeneration in mammalian models of RP (*Arango-Gonzalez et al., 2014*; *Paquet-Durand et al., 2007*; *Power et al., 2020*; *Sancho-Pelluz et al., 2008*; *Tolone et al., 2019*). A summary schematic of the entire screening cascade is provided in *Figure 11*.

Confirmatory tests showed survival effects of lead compounds varied from 9 to 38% (*Figure 2*), which might suggest limited therapeutic potential. However, even small numbers of surviving rod cells can have a significant impact on cone photoreceptor survival (*Guadagni et al., 2015*; *Hartong et al., 2006*; *Punzo et al., 2012*). This effect is mediated by the rod-derived cone viability factor (RdCVF) (*Léveillard et al., 2004*) which stimulates glucose metabolism to promote cone survival (*Aït-Ali et al., 2015*). Therefore, as cone cells provide high-acuity daytime vision, protecting small numbers of rod cell has significant therapeutic potential for RP patients. To test this further, lead compounds will need to be tested for the ability to prolong cone survival and function in rod-cone degeneration models.

Visualization by in vivo confocal imaging showed variations in surviving cell morphologies across lead compounds (*Figure 3* and *Figure 3—figure supplement 2*), suggesting.

variations in the extent of rod cell protection that could impact function. In particular, maintenance of the outer segment, a specialized primary cilium in which phototransduction occurs, is necessary for photoreceptor function. Volumetric quantification of surviving rod cells further suggested lead compounds differ in their capacity to maintain outer segments. Visual function tests will need to be performed to test this possibility and thus to prioritize lead compounds for in vivo testing in mammalian RP models.

To test conservation of neuroprotective effects across species, lead compounds were evaluated in three mouse IRD/RP models. Nine of 11 leads were confirmed as neuroprotectants in primary photoreceptor cell cultures (*Figure 4*). Three of eight leads assayed using *rd1* retinal explant cultures were also validated (*Figure 5*); two being active in both paradigms, DHA and ART. We chose the *rd10* RP model for in vivo testing because it undergoes a slower rate of rod cell loss than *rd1*; spanning from approximately P16 to P35 (*Chang et al., 2007*; *Gargini et al., 2007*). DHA was the most promising compound for these tests as it had shown strong effects across assays and was amenable to a long-term release formulation designed to sustain drug action over weeks to months (PLGA-DHA, *Figure 6A*). In addition, DHA is the active metabolite of artemisinin, another of our cross-species confirmed leads that was shown to have neuroprotective activity in rat models of stress-induced neuronal damage and light-induced photoreceptor degeneration (*Yan et al., 2017*). Unfortunately, we did not observe increased photoreceptor survival in *rd10* retinas injected with PLGA encapsulated DHA. A potential issue that needs to be resolved is the release kinetics of encapsulated DHA in vivo. DHA has poor water solubility, therefore in vitro release kinetics were obtained in the

presence of 0.1% DMSO. However, because intravitreal injection of DMSO can have deleterious effects (*Tsai et al., 2009*) we avoided this potential complication. To assess DHA release in vivo, pharmacokinetic assays such as high-performance liquid chromatography (*Ayalasomayajula and Kompella, 2004*; *Shen et al., 2014*) will be used to quantify lead compound concentrations in the retina for future tests. Another possible explanation is that *rd1* and *rd10* models have differential responses to DHA. This has been reported for the histone-deacetylase inhibitor valproic acid, which shows opposing effects in *rd1* (neuroprotective) and *rd10* (deleterious) mice (*Mitton et al., 2014*) and across four different frog models of RP (*Vent-Schmidt et al., 2017*). In addition valproic acid has produced inconsistent results in clinical trials with RP patients (*Chen et al., 2019*; *Todd and Zelinka, 2017*; *Totan et al., 2017*). These results emphasize the need for a more thorough understanding of IRD and RP disease mechanisms and downstream cell pathways to support the development of both personalized and pan-disease therapeutics.

Numerous IRD/RP-linked mutations have been identified (*Dias et al., 2018*; https://sph.uth.edu/retnet/) implicating an array of disease mechanisms (*Dharmat et al., 2020*). However, cell death pathways common across different IRD/RP patient subpopulations may provide pan-disease targets for neuroprotective therapies. Apoptosis has long been thought to be the mechanism by which rod photoreceptors die in IRD/RP (*Chang et al., 1993*; *Doonan et al., 2003*; *Portera-Cailliau et al., 1994*; *Zeiss et al., 2004*). However, these reports relied on terminal deoxynucleotidyl transferase dUTP nick end labeling (TUNEL), which does not distinguish apoptosis from other types of cell death (*Ansari et al., 1993*; *Charriaut-Marlangue and Ben-Ari, 1995*; *Grasl-Kraupp et al., 1995*; *Dmitrieva and Burg, 2007*; *Kanoh et al., 1999*; *Nishiyama et al., 1996*). More recent evaluations of multiple apoptosis-related markers (e.g. BAX, cytochrome c, caspase-9, cleaved caspase-3) suggest apoptosis occurs in only a minority of RP models (*Arango-Gonzalez et al., 2014*; *Sancho-Pelluz et al., 2008*). Moreover, inhibition of apoptosis does not block cell death in many mouse photoreceptor degeneration models (*Hamann et al., 2009*; *Yoshizawa et al., 2002*), suggesting other pathways may mediate rod and/or cone cell death in retinal degenerative disease.

Recently, non-apoptotic cell death mechanisms have been implicated in IRD/RP. In a comprehensive biochemical analysis of 10 mammalian RP models—involving mutations in *cnga3*, *cngb1*, *pde6a*, *pde6b*, *pde6c*, *prph2*, *rho*, and rpe65—non-apoptotic cell death signatures were found to be common across all models tested (*Arango-Gonzalez et al., 2014*). Conversely, definitive apoptotic markers were found only for the S334ter (*rho*) rat model. Shared features included activation of poly (ADP-ribose) polymerase (PARP), histone deacetylase (HDAC), and calpain, as well as accumulation of cyclic guanosine monophosphate (cGMP) and poly (ADP-ribose) (PAR). For PARP, chemical inhibitors and a knock out line provided further confirmation (*Jiao et al., 2016*; *Paquet-Durand et al., 2007*; *Sahaboglu et al., 2017*; *Sahaboglu et al., 2016*; *Sahaboglu et al., 2010*). Prior reports had suggested the NTR/Mtz system elicits caspase-3 activation and apoptotic cell death (*Chen et al., 2011*). Initially, we had used this as a rationale for pursuing a neuroprotective screen with the *rho: YFP-NTR* line- however, the recent reports outlined above suggested apoptosis may have limited relevance to IRD/RP. Interestingly, when we tested cell death processes implicated in photoreceptor degeneration directly, an inhibitor of apoptosis (BEL) did not promote rod cell survival, whereas inhibition of necroptosis (NEC) and PARP were neuroprotective in our fish RP model (*Figure 7*, *Table 3*). Serendipitously, an exploration of shared lead compound MOA helped to clarify cell death mechanism(s) mediating NTR/Mtz-induced rod cell ablation.

To explore molecular MOA of our lead compounds, we searched bioactivity data from prior HTS and ultra HTS assays (PubChem). The results suggested both shared and independent MOA. The most common shared target was TDP1 (eight of 11 lead compounds; *Table 2*). An initial test confirmed two of three TDP1 inhibitors tested, though neuroprotective activity was relatively weak (~10% survival; *Figure 7*). TDP1 is a DNA repair enzyme that repairs topoisomerase I-induced DNA damage (*Dexheimer et al., 2008*; *El-Khamisy, 2011*). Interestingly, a qHTS cell-based screen of 400,000 compounds for inhibitors of human TDP1 found that all five confirmed compounds actually inhibited PARP activity not TDP1 (*Murai et al., 2014*). This is consistent with findings showing that TDP1 acts in conjunction with PARP1 (*Das et al., 2014*; *Lebedeva et al., 2015*), thus PARP inhibition can indirectly affect TDP1 activity. Combined with the results discussed above, we were motivated to test whether PARP inhibition was protective against NTR/Mtz-induced rod cell death.

All eight PARP inhibitors tested promoted rod cell survival in our fish RP model (*Figure 7* and *Table 3*). CRISPR/Cas9-based knockdown of *parp1* resulted in a statistically significant reduction of

Mtz-induced rod cell death (*Figure 8A*). Moreover, PAR polymer, an indicator of PARP activation, accumulated in Mtz-treated fish (*Figure 8—figure supplement 1C–D*). Collectively, these results suggest PARP activation plays an important role in NTR/Mtz-induced rod cell death. PARP1 overacti-vation initiates a caspase-independent form of DNA damage-induced cell death, termed parthana-tos (*Fan et al., 2017*). Parthanatos has been strongly implicated in the etiology of Parkinson's disease (*Kam et al., 2018*) and in a variety of neurodegenerative conditions as well (*Fan et al., 2017*; *Fatokun et al., 2014*), including RP/IRDs (*Power et al., 2020*). Importantly, PARP is also a key component of the cGMP-dependent cell death pathway which has been linked to photoreceptor degeneration (*Iribarne and Masai, 2017*; *Tolone et al., 2019*; *Power et al., 2020*). Thus, the NTR/ Mtz-mediated cell ablation system may provide a novel approach for modeling neurodegenerative diseases associated with PARP-dependent cell death that is both inducible and titratable.

To investigate cell death more broadly, we also explored the roles of necroptosis, apoptosis and *tdp1* signaling in the zebrafish-inducible RP model. Interestingly, inhibiting necroptosis but not apo-ptosis promoted rod cell survival in *rho:YFP-NTR* fish (*Figure 7* and *Table 3*). Consistent with this, CRISPR/Cas9-based knockdown of *ripk1l* protected rods, but casp3a/3b knockdown did not (*Figure 8*). Necroptosis is primarily associated with secondary cone cell death in RP models (*Murakami et al., 2015*; *Murakami et al., 2012*; *Yang et al., 2017*) but has also been implicated in rod cell death in *IRBP* mutant RP models (*Sato et al., 2013*) and/or may damage photoreceptors indirectly via necroptotic microglia signaling (*Huang et al., 2018*). Combined, these results suggest that necroptosis, PARP1-dependent parthanatos, and/or c-GMP-dependent cell death mediate NTR/Mtz-induced rod cell ablation. The potential relevance of these alternative cell death pathways to heritable photoreceptor degeneration may explain the relatively high rate of validation we observed in cross-species tests of lead compounds.

To evaluate whether lead compound effects were mediated by inhibition of PARP signaling, a subset of lead compounds and a PARP inhibitor control (BMN) were analyzed for survival effects in *parp1* knockdown fish. Among four leads tested that were implicated as *parp1* inhibitors, two (MIC and DHA) were no longer protective when *parp1* expression was suppressed (*Figure 9*), suggesting *parp1* inhibition mediated their neuroprotective effects. PARP1 has also been shown to have a role in stem cell reprogramming (*Chiou et al., 2013*; *Doege et al., 2012*; *Weber et al., 2013*). PARP inhibition might therefore block Müller glia dedifferentiation, diverting injury-induced activity from a regenerative to neuroprotective program (*Bringmann et al., 2009*).

Finally, MOA analyses also implicated additional, potentially complementary, lead targets (*Table 2*). Compounds acting through independent targets/pathways have the potential to produce additive or even synergistic effects. This possibility was confirmed for 10 of 19 paired lead com-pounds tested in *rho:YFP-NTR* fish (*Figure 9*, *Supplementary file 1c*). Analogous tests in mouse photoreceptor cell cultures showed a trend toward enhanced survival effects of paired leads but no fully additive effects (*Figure 4—figure supplement 2*). This discrepancy could be due to cell culture assays lacking in vivo complexity, such as signals requiring intact tissue or interactions between cell types or with other tissues. This result highlights a potential advantage of whole organism pheno-typic drug screening: the ability to identify compounds/conditions that would be missed in more simplified assay systems. That combinatorial assays could be performed efficiently and rapidly also exemplifies key advantages the zebrafish system affords phenotypic drug discovery, for example, versatility and low cost.

In summary, we identified 11 lead compounds promoting rod cell survival in an inducible zebra-fish RP model using a large-scale in vivo phenotypic drug discovery platform. Nine lead compounds were also effective as neuroprotectants in primary mouse photoreceptor cell cultures and three pro-moted photoreceptor survival in *rd1* mouse retinal explants. MOA studies indicated a subset of lead compounds may protect rod cells by inhibiting PARP-dependent cell death pathways and/or necrop-tosis. Combinatorial assays in fish showed additive and even synergistic effects, suggesting some lead compounds target independent neuroprotective pathways (summarized in *Figure 11*). We hypothesize that compounds producing beneficial outcomes across diverse animal disease models likely target highly conserved MOA and may therefore stand an increased likelihood of successfully translating to the clinic. Further, our data suggest polypharmacological targeting of complementary neuroprotective mechanisms has the potential to maximize therapeutic benefits for IRD/RP patients.

# Materials and methods

## Key resources table

| Reagent type (species) or resource | Designation | Source or reference |
|---|---|---|
| Transgenic zebrafish, *Danio rerio* (AB; *roy*) | *Tg(rho:YFP-Eco.NfsB)gmc500; mpv17$^{a9}$* (*rho:YFP-NTR*) | RRID # #:ZFIN_ZDB-GENO-190617–10 |
| Transgenic zebrafish *Danio rerio* (AB; *roy*) | *Tg(rho:GAP-YFP-2A-nfsB_Vv F70A/F108Y)jh405; mpv17$^{a9}$* (*rho:YFP-NTR2.0*) | ZFIN ID pending; Mumm lab |
| *E. coli* | BL21(DE3) Competent Cells | emdmillipore.com: 69450 |
| Mutant mouse *Mus musculus* | *Pde6b$^{rd1}$* (*rd1*) | RRID #:MGI:5902961 |
| Mutant mouse *Mus musculus* (C57BL/6J) | *Pde6b$^{rd10}$* (*rd10*) | RRID # #:IMSR_JAX:004297 |
| Transgenic mouse *Mus musculus* (SJL/J) | QRX-IRES-EGFP | Zack lab |
| Antibody | Anti-Arrestin3a (α-Arr3a; zpr-1) Mouse monoclonal (1:200 dilution) | RRID #:AB_10013803 |
| Antibody | Anti-Rhodopsin (α-Rho; 1d1) Mouse monoclonal - 1:100 dilution | ZDB-ATB-081229–13 |
| Antibody | Ab-4C12 (aka, 4C12) Mouse monoclonal - 1:50: dilution | ZDB-ATB-090506–2 |
| Antibody | Anti-Mouse IgG (H+L), Alexa 647 Goat polyclonal - 1:1000 dilution | RRID #:AB_2338902 |
| Antibody | Anti-Cone Arrestin (mouse) Rabbit polyclonal - 1:1000 dilution | RRID #:AB_1163387 |
| Antibody | Anti-Retinal S antigen (S128) Mouse monoclonal - 1:1000 dilution | RRID #:AB_2747776 |
| Antibody | Anti-PAR (human) Recombinant Antibody - 1:2000 dilution | PMID:30385548 |
| Antibody | Anti-Human IgG (Fab')2 (HRP) Goat polyclonal - 1:10000 dilution | RRID #:AB_1951105 |
| Antibody | Anti-beta-Actin-HRP Mouse monoclonal - 1:20000 dilution | RRID #:AB_262011 |
| Chemical compound | 17-(Allylamino)—17-demethoxygeldanamycin (17-AAG) | CAS #: 75747-14-7 |
| Chemical compound | 17β-Estradiol | CAS #: 50-28-2 |
| Chemical compound | 3-Hydroxybenzylhydrazine dihydrochloride | CAS #: 81012-99-9 |
| Chemical compound | 6,7-Dihydroxyflavone | CAS #: 38183-04-9 |
| Chemical compound | Acacetin | CAS #: 480-44-4 |
| Chemical compound | Ac-DEVD-CHO (AC) | CAS #: 169332-60-9 |
| Chemical compound | AG-14361 | CAS #: 328543-09-5 |
| Chemical compound | Alpha-lipoic acid | CAS #: 1077-28-7 |
| Chemical compound | Alpha-tochopherol | CAS #: 10191-41-0 |
| Chemical compound | Aluminum chloride hexahydrate | CAS #: 7784-13-6 |
| Chemical compound | Artemisinin (ART) | CAS #: 63968-64-9 |
| Chemical compound | Ascorbic acid | CAS #: 50-81-7 |
| Chemical compound | Calcimycin (CAL) | CAS #: 52665-69-7 |
| Chemical compound | Calpastatin peptide | sigmaaldrich.com: SCP0063 |
| Chemical compound | Calpeptin | CAS #: 117591-20-5 |
| Chemical compound | Caspase-3 Inhibitor VII (CASVII) | CAS #: 745046-84-8 |

*Continued on next page*

*Continued*

| Reagent type (species) or resource | Designation | Source or reference |
|---|---|---|
| Chemical compound | Caspase-3/7 Inhibitor I (CASI) | CAS #: 1110670-49-9 |
| Chemical compound | Chloroxine (CHL) | CAS #: 773-76-2 |
| Chemical compound | Ciclopirox olamine (CPO) | CAS #: 41621-49-2 |
| Chemical compound | Clonidine | CAS #: 4205-90-7 |
| Chemical compound | Clopidogrel sulfate | CAS #: 120202-66-6 |
| Chemical compound | Cloxyquine (CLO) | CAS #: 130-16-5 |
| Chemical compound | Compactin | CAS #: 73573-88-3 |
| Chemical compound | Cortexolone (COR) | CAS #: 152-58-9 |
| Chemical compound | D-(-)-Norgestrel | CAS #: 797-63-7 |
| Chemical compound | Danthron | CAS #: 117-10-2 |
| Chemical compound | DAPI dihydrochloride | CAS #: 28718-90-3 |
| Chemical compound | Deltaline | CAS #: 6836-11-9 |
| Chemical compound | Dexamethasone | CAS #: 50-02-2 |
| Chemical compound | Diazepam | CAS #: 439-14-5 |
| Chemical compound | Digoxin | CAS #: 20830-75-5 |
| Chemical compound | Dihydroartemisinin | CAS #: 71939-50-9 |
| Chemical compound | E7449 | CAS #: 1140964-99-3 |
| Chemical compound | Entinostat | CAS #: 209783-80-2 |
| Chemical compound | Escitalopram oxalate | CAS #: 219861-08-2 |
| Chemical compound | Eupatorin | CAS #: 855-96-9 |
| Chemical compound | Hoechst 33342 | CAS #: 23491-52-3 |
| Chemical compound | Hydroquinone | CAS #: 123-31-9 |
| Chemical compound | Indomethacin | CAS #: 53-86-1 |
| Chemical compound | Isopropamide Iodide | CAS #: 71-81-8 |
| Chemical compound | Isoxsuprine hydrochloride | CAS #: 579-56-6 |
| Chemical compound | Lactulose | CAS #: 4618-18-2 |
| Chemical compound | Leucovorin Calcium | CAS #: 1492-18-8 |
| Chemical compound | Levobetaxolol | CAS #: 93221-48-8 |
| Chemical compound | Lobeline sulfate | CAS #: 134-64-5 |
| Chemical compound | Lovastatin | CAS #: 75330-75-5 |
| Chemical compound | Magnesium gluconate | CAS #: 3632-91-5 |
| Chemical compound | Menadione | CAS #: 58-27-5 |
| Chemical compound | Mercaptamine hydrochloride | CAS #: 156-57-0 |
| Chemical compound | Methyl-3,4-dephostatin (MD) | PubChem SID #: 24278575 |
| Chemical compound | Metronidazole (MTZ) | CAS #: 443-48-1 |
| Chemical compound | Miconazole (MIC) | CAS #: 22916-47-8 |
| Chemical compound | MnTBAP | CAS #: 55266-18-7 |
| Chemical compound | Myriocin | CAS #: 35891-70-4 |
| Chemical compound | Myrrh oil | UPC #: 640791683602 |
| Chemical compound | N-Acetyl-L-cysteine (NAC) | CAS #: 616-91-1 |
| Chemical compound | Nalidixic acid | CAS #: 389-08-2 |
| Chemical compound | NCS-382 | CAS #: 520505-01-5 |
| Chemical compound | Necrostatin-1 (NET) | CAS #: 4311-88-0 |

*Continued*

| Reagent type (species) or resource | Designation | Source or reference |
|---|---|---|
| Chemical compound | Niraparib (MK) | CAS #: 1038915-60-4 |
| Chemical compound | NMS-P118 | CAS #: 1262417-51-5 |
| Chemical compound | N-tert-Butyl-α-(2-sulfophenyl)nitrone | CAS #: 73475-11-3 |
| Chemical compound | Olaparib (OLA) | CAS #: 763113-22-0 |
| Chemical compound | Panobinostat | CAS #: 404950-80-7 |
| Chemical compound | Paromomycin (PM) | CAS #: 1263-89-4 |
| Chemical compound | PLGA | CAS #: 26780-50-7 |
| Chemical compound | Pseudoephedrine, (1S,2S)-(+)- | CAS #: 90-82-4 |
| Chemical compound | Retinoic acid (RA) | CAS #: 302-79-4 |
| Chemical compound | Rofecoxib | CAS #: 162011-90-7 |
| Chemical compound | Romidepsin | CAS #: 128517-07-7 |
| Chemical compound | Rosiglitazone | CAS #: 122320-73-4 |
| Chemical compound | Rucaparib (RUC) | CAS #: 283173-50-2 |
| Chemical compound | Streptomycin sulfate | CAS #: 3810-74-0 |
| Chemical compound | Sulindac (SUL) | CAS #: 38194-50-2 |
| Chemical compound | Sunitinib | CAS #: 557795-19-4 |
| Chemical compound | Talazoparib (BMN) | CAS #: 1207456-01-6 |
| Chemical compound | thapsigargin | CAS #: 67526-95-8 |
| Chemical compound | Thiostrepton (ThS) | CAS #: 1393-48-2 |
| Chemical compound | tunicamycin | CAS #: 11089-65-9 |
| Chemical compound | Veliparib (ABT) | CAS #: 912444-00-9 |
| Chemical compound | Vorinostat | CAS #: 149647-78-9 |
| Chemical compound | Warfarin (WAR) | CAS #: 81-81-2 |
| Chemical compound | Xanthurenic acid | CAS #: 59-00-7 |
| Chemical compound | Zinc pyrithione (ZPT) | CAS #: 13463-41-7 |
| Software, algorithm | Software for large-scale in vivo screen data processing (R v3.3.1, R Studio v0.99.903, and the ARQiv2 package) | https://github.com/mummlab/ARQiv2 |
| Software, algorithm | Thermo Scientific HCS Studio Cell Analysis Software | Customized for this project |
| Database | CRISPRScan | https://www.crisprscan.org/ |

## Animal care and use

All animal studies described herein were performed in accordance with both the Association for Research in Vision and Ophthalmology (ARVO) statement on the 'Use of Animals in Ophthalmic and Vision Research' and the National Institutes of Health (NIH) Office of Laboratory Animal Welfare (OLAW) policies regarding studies conducted in vertebrate species. Animal protocols were approved by the Animal Care and Use Committees of the Johns Hopkins University School of Medicine (protocol # FI19M489 and #MO20M253) and Medical University of South Carolina (protocol #2018–00399).

## Fish maintenance and husbandry

Zebrafish were maintained using established temperature and light cycle conditions (28.5°C, 14 hr of light/10 hr of dark). A previously generated zebrafish transgenic line, *Tg(rho:YFP-Eco.NfsB)gmc500* (hereafter, *rho:YFP-NTR*) expresses a fusion protein of enhanced yellow fluorescent protein (YFP) and a bacterial nitroreductase (NTR) enzyme (encoded by the *E. coli nfsB* gene) selectively in rod

photoreceptors (*Walker et al., 2012*). When *rho:YFP-NTR* fish are exposed to the prodrug metronidazole (Mtz), NTR reduces Mtz to a DNA damage-inducing cytotoxic derivative, resulting in the death of NTR-expressing cells (*Chen et al., 2011*; *Curado et al., 2007*). A newly generated transgenic line, *Tg(rho:GAP-YFP-2A-nfsB_Vv F70A/F108Y)jh405* (hereafter *rho:YFP-NTR2.0*) expresses a membrane tagged YFP and an improved version of NTR which enables induction of cell death at much lower concentrations of Mtz (*Sharrock et al., 2020*). These lines were propagated in a pigmentation mutant with reduced iridophore numbers, *roy^{a9}* (*roy*), to facilitate YFP reporter signal detection in vivo. Non-transgenic *roy* fish were used to define reporter signal cutoff values for fluorescent microplate reader assays. For each large-scale drug screening assays, 6000–12,000 eggs were collected by group breeding ~300 adult *rho:YFP-NTR* fish (*White et al., 2016*).

## Immunostaining

*rho:YFP-NTR* larvae were treated with 2.5 mM Mtz or 0.1% DMSO at 5 dpf for 2 days (10 fish per condition), and collected at 7 dpf for immunostaining using previously reported methods (*Unal Eroglu et al., 2018*). Briefly, larvae were fixed in 4% PFA at 4°C overnight, infiltrated with 30% sucrose, and embedded in OCT. Cryosections of 10 μm thickness were made for immunofluorescence staining. Each section was blocked with 1x PBS containing 0.5% Triton X-100% and 5% goat serum for one hour followed by primary antibody staining overnight. The next day, each section was washed and stained with the secondary antibody for 2 hr. All slides were mounted and underwent confocal imaging. The primary antibodies used in this study were zpr-1 (anti-Arrestin3a; 1:200 dilution; from ZIRC), 1d1 (anti-Rhodopsin; 1:50; gift from Dr. James M. Fadool) and 4C12 (uncharacterized rod cell antigen, 1:100 dilution; gift from Dr. James M. Fadool). The secondary antibody used was goat anti-mouse Alexa 647 (1:1000; Invitrogen).

## Optimization of Mtz treatment for rod cell ablation - *rho:YFP-NTR* zebrafish

 *rho:YFP-NTR* larvae were separated into six groups of ~23 larvae per group. Each group was treated with varying Mtz concentrations (10, 5, 2.5, 1.25, 0.625, 0 mM), from 5 to 8 days post-fertilization (dpf). YFP signals were quantified daily by fluorescence microplate reader (TECAN Infinite M1000 PRO; excitation 514 nm, bandwidth 5 nm; emission 538 nm, bandwidth 10 nm) to track changes in rod photoreceptor cell numbers relative to Mtz concentration (*Walker et al., 2012*; *White et al., 2016*). Two technical repeats were performed and results aggregated across experiments by normalizing to non-ablated controls.

## Sample size estimation

To calculate sample size (N) and evaluate assay quality, two 96-well plates of larvae were treated with 2.5 mM Mtz/0.1% DMSO (ablated '+Mtz' control) or 0.1% DMSO only (non-ablated '-Mtz' control) from 5 to 7 dpf (196 fish per condition). YFP signals were then quantified at seven dpf (due to the large sample size, a single technical replicate was performed). Power calculations were used to determine sample sizes across a range of error rates and effect sizes (*White et al., 2016*). This analysis suggested a sample size of nine per condition tested would sufficiently minimize false-discoveries (i.e. type I and type II error rates of 0.05 and 0.05, respectively) and account for drug effects reaching 50% of signal window of the positive (non-ablated) control. However, to account for plating errors and other confounding variables, we chose to increase the sample size to 16.

## Primary drug screening assay - *rho:YFP-NTR* zebrafish

A schematic of the primary screening process is presented in *Figure 1C*. For the primary screen, 2934 compounds from John Hopkins Drug Library (JHDL) were screened across six concentrations (4 μM-125 nM using a twofold dilution series). The JHDL consists of ~2200 drugs approved for use in humans (e.g. FDA approved) with the remainder approved for clinical trials (*Chong et al., 2006*). The ARQiv screening process has been detailed previously (*White et al., 2016*) and was adapted here for large-scale quantification of YFP-expressing rod photoreceptors (*Walker et al., 2012*; *White et al., 2016*). *rho:YFP-NTR* embryos were collected and raised in zebrafish E3 embryo media (5 mM NaCl; 0.17 mM KCl; 0.33 mM CaCl; 0.33 mM MgSO$_4$). At 16 hr post fertilization (hpf), N-phenylthiourea (PTU) was added to E3 media (E3/PTU) at a final concentration of 0.2 mM to

promote ocular transparency by inhibiting melanosome maturation in the retinal pigment epithelium. At 4 dpf, visual screens were performed to remove larvae with abnormal morphology or low YFP expression levels. Stock drug and DMSO (negative control) solutions were automatically dispensed and diluted across a 96-well plate containing E3/PTU using a robotic liquid handling system (Hudson Robotics). At 5 dpf, a COPAS-XL (Complex Object Parametric Analyzer and Sorter, Union Biometrica) was used to dispense single larvae into individual wells containing either drug or DMSO; the final DMSO concentration was 0.1% across all conditions. After a 4 hr pre-exposure to test drugs or DMSO alone, larvae were treated with 2.5 mM Mtz to induce rod photoreceptor death. Larvae were maintained under these conditions for 2 days until 7n dpf and then anesthetized with clove oil (50 ppm final concentration); YFP signals were measured as described above. Larvae exposed to 2.5 mM Mtz/0.1% DMSO without any tested drug served as controls ('+Mtz') for maximal rod cell ablation. Larvae treated solely with 0.1% DMSO served as non-ablated controls ('-Mtz') to calculate maximal YFP signal levels. Non-transgenic larvae were used to establish a signal cutoff value, as previously described (*White et al., 2016*). A customized R code program was applied for real-time data analysis of compound performance relative to controls, including dose-response curves and strictly standardized mean difference (SSMD) scores (https://github.com/mummlab/ARQiv2; *Ding and Zhang, 2021*). Compound concentrations producing a SSMD score $\geq 1$ were considered potential 'hits' and evaluated visually using fluorescence microscopy to eliminate non-specific fluorescence; e.g., dead larvae or autofluorescent compounds.

## Secondary confirmation assay - rho:YFP-NTR zebrafish

After the initial screen, 42 top-performing 'hit' compounds were selected for confirmatory and orthogonal assays. Hit drugs were obtained from new sources and tested across a wider range of concentrations (fivefold dilution series, eight total concentrations, n=30 fish/condition). Based on the toxicity profile of each drug, the starting concentration was either 100, 10, or 1 µM. For validation assays, YFP signals were measured by fluorescence microplate reader with at least three experimental repeats conducted and SSMD scores calculated. All experimental results were normalized and pooled to calculate effect sizes, confidence intervals, sample sizes (N) and p-values. p Values were calculated using Student's t-test followed by Bonferroni correction for multiple comparisons resulting in an adjusted significance level of 0.005 ($\alpha$=0.005).

## Confocal imaging - rho:YFP-NTR and rho:YFP-NTR2.0 zebrafish

For in vivo confocal imaging assays, five dpf *rho:YFP-NTR* larvae were treated with drug plus 2.5 mM Mtz/0.1% DMSO (ablated +Mtz controls), or 0.1% DMSO (non-ablated -Mtz controls) for 48 hr and processed for intravital imaging at 7 dpf. Similar treatment strategy was applied to five dpf *rho:YFP-NTR2.0* larvae except the Mtz treatment was reduced to 10 µM. All compounds were tested at the maximal effective concentration in 0.1% DMSO. Larvae from each group were anesthetized in 0.016% tricaine and embedded on their sides in 1% low melt agarose gel. An Olympus Fluoview FV1000 confocal microscope with a 20x water immersion objective (0.95 NA) was used to collect 30–40 z-stack images of YFP-expressing rod cells across the whole retina at 4 µm intervals. These image slices were stacked into a single maximal intensity projection image using Image J. To provide greater detail of rod photoreceptor morphology, a region in the dorsal-nasal quadrant was imaged using a 60× water immersion objective (1.10 NA) at 4.18 µm intervals. Three contiguous image slices were then stacked into a single maximal intensity projection image using Image J. As a means of assessing rod photoreceptor survival, intravital confocal retinal images were collected of YFP-expressing rod cells per condition and processed for volumetric quantification of YFP signal volume using automated image analysis software (Imaris 3.9.1; see *White et al., 2017*). Briefly, YFP-expressing rod cells volumes were automatically rendered using the same parameters across all treatment groups with the sum of all volumes used to estimate rod photoreceptor numbers per each retina. For images from *rho:YFP-NTR2.0* larvae, the sum of rod cell YFP volumes in each retina was normalized to the number of YFP-expressing cells to estimate the average rod photoreceptor cell size per each condition. The average number of cells quantified per retina was 18.52, with sample sizes ranging from 9 to 19 fish per condition.

## In vitro nitroreductase inhibition assay

The eleven lead compounds were tested to eliminate false-positives that directly inhibited *E. coli* NTR enzymatic activity (from the *nfsB_Ec* gene) using an established anticancer prodrug (CB1954) reduction assay (*Prosser et al., 2010*). The highest soluble concentration of the test drug (up to 300 µM maximum) or 0.1% DMSO (rate control) was added to a master mix of 250 µM CB1954/1 µM NTR/250 µM NADPH/10 mM Tris (pH 7.0) to determine any inhibitory effect of the compounds on NTR-mediated reduction of CB1954. Reactions were conducted in 100 µl volume in a 96-well plate format. CB1954 reduction kinetics were assayed at 420 nm unless the test compound exhibited confounding absorbance at 420 nm, in which case NADPH depletion was monitored at 340 nm. The reaction rates for CB1954 reduction in the presence of tested compounds were compared to the DMSO control. Compounds with NTR activity <75% of controls were deemed potentially inhibitory and $IC_{50}$ values (the concentration required to restrict NTR activity to 50% of the DMSO control) determined.

Compounds were also evaluated for the ability to inhibit NTR-mediated reduction of Mtz. For this assay, the highest soluble concentration of the test drug (up to 300 µM maximum) or DMSO (rate control) was added to a master mix of 500 µM Mtz/17 µM NTR/100 µM NADPH/0.55 µM glucose dehydrogenase/5 mM glucose/50 mM sodium phosphate buffer (pH 7.0; the glucose dehydrogenase used for cofactor regeneration, maintaining a steady-state of NADPH to allow Mtz reduction to be monitored directly at 340 nm without interference from NADPH oxidation). Reactions were conducted in 100 µl volume and assayed at 340 nm using 1 mm quartz cuvettes (*Rich et al., 2018*). $IC_{50}$ values for potentially inhibitory compounds were measured as with CB1954. All kinetic assays were conducted in triplicate with a single purified batch of NfsB_Ec protein.

## Rod photoreceptor Neogenesis assay - rho:YFP-NTR zebrafish

To evaluate whether lead compounds promoted rod photoreceptor development, *rho:YFP-NTR* larvae were handled as described for the primary screen with the following exceptions: at five dpf, larvae were exposed solely to test compounds; larvae were not treated with Mtz. YFP reporter signals were quantified by fluorescence microplate reader at 7 dpf and compared to non-ablated '-Mtz' controls treated only with 0.1% DMSO. Sample sizes ranged from 58 to 88 across at least two experimental repeats.

## Rod photoreceptor regeneration assay - rho:YFP-NTR zebrafish

To determine whether lead compounds stimulated retinal regeneration, *rho:YFP-NTR* larvae were handled as described for the primary screen with the following exceptions: at 5 dpf, larvae were incubated with either 10 mM Mtz or 0.1% DMSO for 24 hr. At 6 dpf, larvae were then placed in new 0.1% DMSO/E3/PTU media containing test compounds (or DMSO alone) for 3 days. YFP signal intensity was measured at 9 dpf. Sample sizes ranged from 31 to 83 across at least two experimental repeats.

## Primary mouse retinal photoreceptor cell culture assays

A transgenic mouse line with an IRES-GFP cassette integrated at the QRX locus (QRX-IRES-EGFP) has GFP reporter expression restricted to retinal photoreceptors, and was used here to isolate primary photoreceptor cells. Mice were housed with 12 hr light/12 hr dark cycles, at 22°C, 30–70% relative humidity and food/water ad libitum. Primary mouse retinal photoreceptor cells were isolated and prepared for culture as previously described (*Fuller et al., 2014*). Briefly, murine retinas were isolated at postnatal day four (P4). Retinal tissue was dissociated into a single-cell suspension by incubating whole tissue in activated papain in Hibernate-E without Calcium (BrainBits) for 15 min at 37°C. Cells were resuspended in culture media (Neurobasal, 2% B-27, 0.5 mM L-Glutamine and 1x final Penicillin/streptomycin; all Life Technologies) and seeded onto poly-D-lysine coated 384 well tissue culture plates. To test lead compounds for survival effects, thapsigargin and tunicamycin (Sigma) and were used as stressor compounds to induce photoreceptor cell death. Stressors and test compounds were added at the time of primary cell seeding. Eleven lead compounds were tested at seven concentrations across a threefold dilution series (from 30 µM to 40 nM). For paired tests, each compound was applied at a single concentration, WAR (3.33 µM), CPO (0.04 µM), DHA (1.11 µM), ART (10 µM), ZPT (0.12 µM), CAL (0.04 µM), COR (3.33 µM), CHL (3.33 µM), SUL (3.33 µM), MIC (10

µM), and CLO (1.11 µM). After a 48 hr treatment, cells were stained with Hoescht and ethidium homodimer; nine field images were acquired via an automated imager (Cellomics Vti; ThermoFisher) using a 20x objective and images analyzed. Photoreceptor number and the percent of photoreceptor to retinal cell population per well was determined by automated quantification of the number of live Hoechst 33342-stained, GFP-expressing cells using a custom algorithm (Neuronal Profiling software package; ThermoFisher). A minimum of two experimental repeats was performed for each test with individual assays consisting of six biological replicates per condition tested.

## Retinal explant assay - Pde6b$^{rd1}$ (rd1) mouse

Mice were generated from *retinal degeneration 1* (*Pde6b$^{rd1}$*, hereafter *rd1*) breeding pairs (gift from Dr. Debora Farber; UCLA) and housed under a 12:12 light:dark cycle with access to food and water ad libitum. All chemicals used for organ cultures were tissue culture grade and purchased from Invitrogen unless otherwise noted. Cultures of retina with attached retinal pigment epithelium (RPE) were grown according to published protocols (*Ogilvie et al., 1999*; *Pinzón-Duarte et al., 2000*; *Rohrer and Ogilvie, 2003*) with modifications (*Bandyopadhyay and Rohrer, 2010*). P10 pups were decapitated and heads were rinsed in 70% ethanol; whole eyes were dissected and placed in ice-cold Hanks balanced salt solution plus glucose (6.5 g/L). Eyes were first incubated in 1 mL of high-glucose Hanks balanced salt solution/0.5 mg/ml proteinase K (37°C, 7 min) and then in Neurobasal medium (Life Technologies) plus 10% fetal calf serum to stop enzymatic activity. The retina with attached RPE was dissected free from the choroid and sclera after removing the anterior chamber, lens and vitreous. Relaxing cuts were made to flatten the tissue prior to transfer to the upper compartment of a Costar Transwell chamber (RPE layer faced-down) using a drop of Neurobasal medium. Neurobasal media with B-27 supplement (Life Technologies) was placed in the lower compartment. For each of the cohorts, three to fourindividual P10 retina/RPE explants were placed in culture. The cultures were kept in an incubator (5% $CO_2$, balanced air, 100% humidity, 37°C) and the lower compartment media changed every 2 days (neither antimitotics nor antibiotics were required). Test compounds were added to the culture media and refreshed every 2 days for 11 days. At completion, explants were fixed in 4% PFA, sectioned 14 µm thick, and stained with toluidine blue (*Bandyopadhyay and Rohrer, 2010*). For each culture, 10 counts of rows of photoreceptors were performed in the center of the explant; the average of this cell counts provides the mean for each retina, with the average of all retinas providing the mean ± SEM per culture condition. Cell counts were compared by repeated measure ANOVA to assess dose-response treatment effects followed by a Bonferroni posthoc analysis to determine if differences between control and lead compounds compared across the entire dose range were statistically significant. An ANOVA across groups but within a particular dose, using a Bonferroni/Dunn test to reduce type I errors, was used to determine if lead compounds promoted a statistically significantly increase in cell numbers compared to control media at low, middle, and/or high concentrations.

## Long-release formulation of DHA

PLGA-DHA microparticles were prepared using a single emulsion solvent evaporation method. Briefly, 200 mg PLGA (Resomer RG 502 hr, Poly(D),L-lactide-co-glycolide; Evonik Corporation) was dissolved in 1 mL of dichloromethane (DCM, Sigma-Aldrich), and mixing with 40 mg DHA (TCI, Tokyo, Japan) dissolved in 0.125 ml dimethyl sulfoxide (DMSO, Sigma-Aldrich). The mixture was homogenized (L4RT, Silverson Machines) at 7000 RPM for 1 min. The homogenized mixture was then poured into a solution containing 1% polyvinyl alcohol (25 kDa, 88% hydrolys, Polysciences, Warrington, PA) in water under continuous stirring. Particles were hardened by allowing solvent to evaporate while stirring at room temperature for 2 hr. Particles were collected via centrifugation (International Equipment Co) at 1000 x g for 5 min, and washed three times with HyPure cell culture grade water (endotoxin-free, HyClone, Logan, UT) and re-collected by centrifugation three times. The washed particles were then lyophilized and stored frozen until use. Microparticles were resuspended at the desired concentration prior to injection in a sodium hyaluronate solution (Healon) diluted fivefold with endotoxin-free HyPure water at the desired concentration prior to injection.

## Characterization of PLGA-DHA microparticles

Particle size distribution was determined using a Coulter Multisizer 4 (Beckman Coulter, Inc, Miami, FL). Particles were resuspended in double distilled water and added dropwise to 100 ml of ISOTON II solution until the coincidence of the particles was between 8% and 10%. At least 100,000 particles were sized for each batch of particles to determine the mean particle size and size distribution. To determine the drug loading, microparticles were dissolved in DMSO and the total drug content was calculated by measuring the UV absorbance at 289 nm after react with NaOH in triplicate (C.-S. Lai et al., 2007a). The average microparticle size was 14.2 ± 1.9 µm and the DHA loading was 20.3% ± 0.7 (w/w) across three experimental replicates.

## PLGA-DHA release kinetics in vitro

Release kinetics were obtained by resuspending microparticles in 1 ml phosphate buffered saline (PBS containing 0.1% DMSO, pH 7.4) and incubating at 37°C on a platform shaker (140 RPM). Supernatant was collected at predetermined intervals by centrifugation at 2000 x g for 5 min. Drug-containing supernatant was collected and particles were resuspended in 1 ml of fresh 1xPBS containing 0.1% DMSO. DHA concentration in the collected supernatant was assayed via absorbance at 289 nm in triplicate for each sample (*Figure 4—figure supplement 1*).

## In vivo mouse retinal assay - Pde6b$^{rd10}$ (rd10) mouse model

B6.CXB1-*Pde6b$^{rd10}$*/J animals (Jackson Laboratories stock #004297) were maintained as homozygotes. On P14, animals were anesthetized with ketamine/xylazine (115 mg/kg and 5 mg/kg, respectively), and placed on a heating pad to maintain body temperature throughout injection and recovery. Glass needles were pulled and a bore size of approximately 75 µm was made to allow for the movement of the microspheres into and out of the needle. A PLI-100 picospritzer (Harvard Apparatus) was used for intravitreal injections with settings of 350 ms injection time at 30 psi, which yielded the injection of approximately 500 nL. One eye was randomly selected for injection with vehicle and the other was used for microsphere injections. Following injection, GenTeal gel (Novartis) was applied to the eyes to prevent corneal drying. Animals were monitored until they were awake and moving normally. A total of 17 mice was used across three technical repeats.

Eighteen days post-injection, animals were anesthetized with isoflurane and decapitated; the eyes were removed and placed in 4% PFA for 30 min. After removing the cornea and lens, the eye cups were placed back into fixative for 2 hr on ice. After thorough washing with 1× PBS, eyecups were soaked in 15% sucrose (w/v) in 1× PBS for 3 hr, then incubated in 30% sucrose (w/v)/1× PBS overnight. Samples were embedded in OCT compound (TissueTek) and frozen on dry ice. Cryosections of 12–14 µm thickness were cut on a Leica CM3050S cryostat and were mounted on Superfrost Plus slides (Fisher Scientific). Slides were rehydrated with 1× PBS and then incubated in blocking buffer (10% Normal donkey serum/ 0.3% Triton X-100/1× PBS) for 2 hr, and were then incubated for 30 hr at 4°C with primary antibodies diluted in blocking buffer. Sections were washed in 1× PBS and incubated for 2 hr at room temperature with secondary antibodies diluted in blocking buffer. Following additional washes, sections were incubated with DAPI and mounted with FluoroGel (Electron Microscopy Sciences).

Sections were examined using a Zeiss Axioplan two epifluorescence microscope fitted with an Axioscope camera. Following acquisition with a 40× air objective, images were analyzed in ImageJ using the measure tool to examine Outer Nuclear Layer (ONL) thickness. Four to 5 measurements were taken at 75 µm intervals for 200–500 µm both superior and inferior from the optic nerve head. The measured values from multiple sections were averaged for each eye. Data were analyzed and statistical tests were performed in GraphPad Prism 7. An Olympus FV1000 confocal microscope was also used to image sections stained with multiple dyes.

## Shared MOA and cell death pathway inhibitor assays – rho:YFP-NTR zebrafish

To evaluate *tdp1* and *parp1* as a potential shared target across multiple leads, as well as cell death pathways implicated in photoreceptor degeneration, eight PARP inhibitors, one necroptosis inhibitor, three apoptosis inhibitors, and three Tdp1 inhibitors were selected for testing using the confirmation screening protocol (*Figure 2A*). Based on the toxicity profile of each inhibitor, upper concentrations

were adjusted to 64, 32, or 8 µM and a twofold dilution series was used to test a total of 10 concentrations for rod cell survival effects in *rho:YFP-NTR* larvae. Sample sizes varied from 27 to 173 across two to six experimental replicates for each tested compound. YFP signal levels were measured by fluorescence microplate reader (i.e. ARQiv assay). Results across conditions were normalized to non-ablated controls and pooled across experimental repeats to calculate effect sizes, confidence intervals, and *p*-values. *P* values were calculated by performing Student's t-tests. Bonferroni correction for multiple comparisons resulted in an adjusted significance level of 0.003 ($\alpha$=0.003). Note: assessments of TDP1 inhibitors followed the fivefold dilution series protocol used for secondary confirmation assays described above. For paired inhibitor tests, three concentrations of BMN (parp inhibitor) and three concentrations of NEC (necroptosis inhibitor) were combined. YFP quantification and statistical analysis of rod cell survival in Mtz-treated *rho:YFP-NTR* fish were performed as above.

## CRISPR/Cas9-based gene knockdown

Five genes, *parp1*, *ripk1l*, *casp3a*, *casp3b*, *tdp1*, were selected for CRISPR/Cas9-based knockdown in *rho:YFP-NTR* zebrafish using an established 'redundant targeting' methodology (*Wu et al., 2018*). Four sgRNAs were designed to target each gene. The oligonucleotide for generating each sgRNA was either those suggested by *Wu et al., 2018*, Table S2 or designed using CRISPRscan (https://www.crisprscan.org, *Supplementary file 3*; *Moreno-Mateos et al., 2015*). Preparation of sgRNAs followed the published method (*Wu et al., 2018*). Briefly, four sgRNAs of each gene were in vitro transcribed using the HiScribe T7 High Yield RNA Synthesis kit (New England BioLabs) and purified using the NEB Monarch RNA Cleanup kit. A mixture of four sgRNAs (1 ng in total) and Cas9 protein (2.5 µM, IDT) was injected into *rho:YFP-NTR* embryos at the one-cell stage for targeting *parp1*, *ripk1l* and *tdp1*. Four sgRNAs each for *casp3a* and *casp3b* were pooled to co-target both genes. Injected and uninjected embryos were treated ±Mtz (2.5 mM) at 5 dpf. Rod cell survival was evaluated by fluorescence microplate reader-based quantification of YFP at 6 dpf. Student's *t*-test was used to compare rod survival between injected and uninjected (control) fish. Each gene knockdown experiment was conducted in triplicate or quadruplicate. A subset of lead compounds (and a PARP inhibitor control, BMN) was also tested in *parp1* knockdown fish to determine if Parp1 inhibition was required for survival effects. Leads were applied to larvae one hour before Mtz administration and YFP quantification performed as above. Each drug test was conducted in duplicate or triplicate. To calculate p-values relative to +Mtz controls, Student's *t*-tests were performed. Bonferroni correction for multiple comparisons resulted in an adjusted significance level of 0.006 ($\alpha$=0.006).

## Quantitative PCR

To quantify gene expression levels in *rho:YFP-NTR* larvae ±CRISPR/Cas9-based gene knockdown, 16–24 larvae from each condition were pooled for mRNA extraction at six dpf. RNA was purified using NEB Monarch RNA Cleanup kit and reverse transcribed to cDNA using qScript cDNA synthesis kit (QuantaBio) according to the instruction from the manufacturer. Quantitative PCR was conducted using designed primers (*Supplementary file 3*) and PowerUp SYBR Green Master Mix (ABI) in QuantaStudio (ABI). Delta delta CT analysis was performed to calculate relative fold change in gene expression levels between knockdown larvae and controls. Each experiment was performed in triplicate or quadruplicate.

## Western blot

To further evaluate Parp-dependent signaling downstream of Mtz treatments in *rho:YFP-NTR* larvae, PAR accumulation, a vetted measure of Parp activity, was quantified by western blot following the published method (*Kam et al., 2018*). Briefly, ~30 *rho:YFP-NTR* fish were treated ±2.5 mM Mtz at five dpf were collected at 3, 6, 12, 24, 48 hr post treatment (hpt). Larvae were homogenized and lysed using RIPA buffer (Sigma) supplemented with protease inhibitors (Roche). Protein concentrations were quantified using Pierce BCA kit (ThermoScientific) following the manufacturer's instruction. Protein samples were separated on Tris-Glycine gel and then transferred onto nitrocellulose membranes. The membrane was blocked with 5% non-fat milk in TBST (Tris-buffered saline with 0.1% Tween 20) at room temperature for 1 hr and then blotted with the primary antibody, anti-human-PAR (1:2000, Dowson lab reagent), at 4°C overnight. After rinsing in TBST for three times, the secondary antibody, anti-Human IgG-HRP (1:10,000, Abcam) was applied for 1 hr at room

temperature. Anti-beta-actin-HRP antibody (1:20,000) was used as a loading control. Immunolabeled bands were visualized by ECL substrate and quantified in Fiji. 24 hr post-Mtz PAR accumulation assays were performed in triplicate. Mann-Whitney U test was performed and a p-value of $\leq 0.05$ used to indicate statistical significance.

## Paired drug testing in zebrafish

To test for additive neuroprotective effects in zebrafish, seven lead compounds were tested at optimal concentrations either individually or in pairs using the primary screen protocol. All experimental results were normalized to controls and pooled per compound tested to calculate effect sizes, confidence intervals, and p-values using Student's t-test. Bonferroni correction for multiple comparisons resulted in an adjusted significance level of 0.002 ($\alpha$=0.002). Sample sizes for each paired condition varied from 86 to 160 across at least four experimental replicates.

## Data analysis and statistics

For the primary screening, under the R environment, a customized ARQiv data analysis package was used to calculate sample size (n), quality control strictly standardized mean difference (SSMD) and SSMD scores as previously described (*White et al., 2016*). The following results of each drug were derived: (1) a plot of signal to background ratio at all tested concentrations, (2) a plot and a table of SSMD scores, and (3) a signal intensity heat map of each drug plate (96-well plate view). To combine and analyze the data from different experiments, data normalization was conducted by $(S_i - \bar{X}_{neg})/(\bar{X}_{Pos} - \bar{X}_{neg})$. $S_i$ is the signal of $i^{th}$ reading, $\bar{X}_{Pos}$ is mean of positive controls, and $\bar{X}_{Neg}$ is mean of negative controls. To compare between experimental conditions and controls, Student's t-tests were performed followed by Bonferroni correction for multiple comparisons ($\alpha/n$). Relative effects, 95% confidence intervals and p-values were calculated. For mouse photoreceptor cell culture assays and volumetric confocal image analyses, adjusted p-values were calculated by performing Mann-Whitney *U* tests followed by false discovery rate correction for multiple comparisons.

## Acknowledgements

This project was supported by a Wynn-Gund TRAP award from the Foundation Fighting Blindness (JSM). Additional funding was provided by the Wilmer core grant for vision research, microscopy and imaging core module (NIH P30-EY001765), the National Institutes of Health (NIH) R01EY019320 (BR), the Department of Veterans Affairs RX000444 and BX003050 (BR), and the South Carolina SmartState Endowment (BR). Assembly of the JHDL was supported by the Flight Attendant Medical Research Institute and ITCR (JOL). We thank Dr. James M Fadool (Florida University) for providing the 1d1 and 4C12 monoclonal antibodies, and Dr. Debora B Farber (University of California, Los Angeles) for providing *Pde6b^{rd10}* mice.

## Additional information

### Competing interests

Liyun Zhang: has filed a provisional patent for the discoveries described herein. Jeff S Mumm: holds patents for the NTR inducible cell ablation system (US #7,514,595) and uses thereof (US #8,071,838 and US#8431768). Has filed a provisional patent for the discoveries described herein. All other authors have no commercial relationships. The other authors declare that no competing interests exist.

### Funding

| Funder | Grant reference number | Author |
| --- | --- | --- |
| Foundation Fighting Blindness | TA-NMT-0614-0643-JHU-WG | Jeff S Mumm |
| National Institutes of Health | P30EY001765 | Donald J Zack |
| National Institutes of Health | R01EY019320 | Baerbel Rohrer |

| Department of Veterans Affairs | RX000444 | Baerbel Rohrer |
| University of South Carolina | South Carolina Smart State Endowment | Baerbel Rohrer |
| Flight Attendant Medical Research Institute | | Jun O Liu |
| Department of Veterans Affairs | BX003050 | Baerbel Rohrer |
| National Institutes of Health | R01OD020376 | Jeff S Mumm |
| National Institutes of Health | R01EY022810 | Jeff S Mumm |

The funders had no role in study design, data collection and interpretation, or the decision to submit the work for publication.

## Author contributions
Liyun Zhang, Conceptualization, Data curation, Formal analysis, Supervision, Validation, Investigation, Visualization, Methodology, Writing - original draft, Writing - review and editing; Conan Chen, Formal analysis, Validation, Investigation, Visualization; Jie Fu, Formal analysis, Supervision, Validation, Methodology, Writing - original draft, Writing - review and editing; Brendan Lilley, Cynthia Berlinicke, Conceptualization, Formal analysis, Investigation, Methodology, Writing - original draft, Writing - review and editing; Baranda Hansen, Investigation; Ding Ding, Data curation, Software, Formal analysis, Visualization, Writing - review and editing; Guohua Wang, Software, Formal analysis, Investigation; Tao Wang, Ying Ye, Data curation, Formal analysis, Validation; Daniel Shou, Data curation, Formal analysis; Timothy Mulligan, Jun O Liu, Resources, Writing - review and editing; Kevin Emmerich, Formal analysis, Investigation, Methodology, Writing - review and editing; Meera T Saxena, Conceptualization, Resources, Methodology, Project administration, Writing - review and editing; Kelsi R Hall, Abigail V Sharrock, Validation, Investigation, Visualization, Methodology, Writing - review and editing; Carlene Brandon, Validation, Investigation; Hyejin Park, Tae-In Kam, Formal analysis, Investigation, Methodology; Valina L Dawson, Ted M Dawson, Supervision, Funding acquisition, Project administration, Writing - review and editing; Joong Sup Shim, Resources; Justin Hanes, Supervision, Project administration, Writing - review and editing; Hongkai Ji, Formal analysis, Supervision, Writing - original draft; Jiang Qian, Supervision, Investigation; David F Ackerley, Resources, Supervision, Methodology, Writing - review and editing; Baerbel Rohrer, Resources, Data curation, Formal analysis, Supervision, Funding acquisition, Validation, Visualization, Methodology, Project administration, Writing - review and editing; Donald J Zack, Conceptualization, Resources, Data curation, Formal analysis, Supervision, Funding acquisition, Visualization, Methodology, Project administration, Writing - review and editing; Jeff S Mumm, Conceptualization, Resources, Data curation, Formal analysis, Supervision, Funding acquisition, Visualization, Methodology, Writing - original draft, Project administration, Writing - review and editing

## Author ORCIDs
Valina L Dawson (ID) https://orcid.org/0000-0002-2915-3970
Ted M Dawson (ID) https://orcid.org/0000-0002-6459-0893
Jun O Liu (ID) http://orcid.org/0000-0003-3842-9841
David F Ackerley (ID) https://orcid.org/0000-0002-6188-9902
Jeff S Mumm (ID) https://orcid.org/0000-0002-2575-287X

## Ethics
Animal experimentation: All animal studies described herein were performed in accordance with both the Association for Research in Vision and Ophthalmology (ARVO) statement on the "Use of Animals in Ophthalmic and Vision Research" and the National Institutes of Health (NIH) Office of Laboratory Animal Welfare (OLAW) policies regarding studies conducted in vertebrate species. Animal protocols were approved by the Animal Care and Use Committees of the Johns Hopkins University School of Medicine (protocol # FI19M489 and #MO20M253) and Medical University of South Carolina (protocol #2018-00399).

Decision letter and Author response
Decision letter https://doi.org/10.7554/eLife.57245.sa1
Author response https://doi.org/10.7554/eLife.57245.sa2

# Additional files

## Supplementary files

• **Supplementary file 1.** Statistics of Mtz titration assay. (a) Statistical summary of Mtz titration assay in rho:YFP-NTR zebrafish larvae - *Figure 1B*. Survival effects (normalized YFP, %), 95% confidence intervals, p-values, and sample sizes (N) for each condition at 7 dpf. Student's t-test was used to calculate p-values for each condition relative non-ablated controls (0 mM Mtz). Bonferroni correction for multiple comparisons resulted in an adjusted alpha level of 0.01 ($\alpha=0.01$). Two experimental repeats were performed for each condition and data pooled across replicates (*Figure 1—source data 1*). (b) Paired compound p-values relative to all control conditions in *Figure 7—figure supplement 1*. Student's t-test was used to calculate p-values for each paired condition relative to ablated controls (+Mtz), BMN alone control and NEC alone control for all paired conditions. Bonferroni correction for multiple comparisons resulted in an adjusted significance level of 0.003 ($\alpha=0.003$). Survival effects as shown in *Figure 7—figure supplement 1* are provided for context (*Figure 7—figure supplement 1—source data 1*). Inhibitor abbreviations: BMN, talazoparib; NEC, necrostatin-1. Other abbreviations: CI, confidence interval; Mtz, Metronidazole; NA, not applicable. (c) Paired compound p-values relative to all control conditions in *Figure 10*. Student's t-test was used to calculate p-values for each paired condition relative to ablated controls (+Mtz) and relevant individual compound controls (Cmpd A, top paired compound; Cmpd B, bottom paired compound). Bonferroni correction for multiple comparisons to +Mtz control resulted in an adjusted significance level of 0.002 ($\alpha=0.002$); significance level for comparisons to individual compound controls was 0.05 ($\alpha=0.05$). Survival effects as shown in *Figure 10* are provided for context (*Figure 10—source data 1*). Lead compound abbreviations: WAR, Warfarin; CLO, Cloxyquin; CPO, Ciclopirox olamine; MIC, Miconazole; ZPT, Zinc pyrithione; DHA, Dihydroartemisinin; CHL, Chloroxine; CAL, Calcimycin; SUL, Sulindac; ART, Artemesinin; COR, Cortexolone; POS, positive control. Other abbreviations: CI, confidence interval; Mtz, Metronidazole.

• **Supplementary file 2.** Previously implicated neuroprotectants. (a) Compounds tested as positive controls. List of 17 compounds previously reported as neuroprotectants in RP models tested for survival effects in *rho:YFP-NTR* zebrafish larvae using the primary screening protocol. (b) List of eliminated compounds. Compounds that were autofluorescent (precluding YFP signal detection) or lethal at the concentrations tested (10 mM to 0.625 mM). (c) List of 113 hit compounds. Hit compounds producing a SSMD score ≥1 in the primary screen ordered according to SSMD score. Drug names, concentrations producing SSMD ≥1, SSMD scores, SSMD effect types, and whether a dose-dependent trend was observed or not are shown. Yellow highlighted drugs were selected for confirmation testing. ''•'' denotes confirmed lead compounds (source data). (d) On-label MOA for 113 hit compounds. Implicated MOA categories and subcategories are listed in order from most common to least common. The number of compounds per each category/subcategory are provided in the parentheses and compound names are listed.

• **Supplementary file 3.** Oligonucleotides used for sgRNA synthesis (gene knockdown) and qPCR primers. Abbreviations: *parp1*, poly (ADP-ribose) polymerase 1; *ripk1l*, receptor (TNFRSF)-interacting serine-threonine kinase 1, like; *casp3a*: caspase 3, apoptosis-related cysteine peptidase a; *casp3b*: caspase 3, apoptosis-related cysteine peptidase b; *tdp1*, tyrosyl-DNA phosphodiesterase 1; *actb1*: actin, beta 1; *rplp0*: ribosomal protein, large, P0.

## Data availability

Source data files have been uploaded. The oligonucleotide for generating each sgRNA was either those suggested by Wu et al., 2018, Table S2 or designed using CRISPRscan (https://www.crisprscan.org, Supplementary file 3; Moreno-Mateos et al., 2015).

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
