## [Decision Letter]

**Acceptance summary:**

New therapeutic targets are needed for retinitis pigmentosa, a leading cause of blindness. In this elegant study, a large scale drug screen anchored to a multi species validation platform identified several compounds that promote photoreceptor cell survival in RP models. Comprehensive follow up pharmaco-genetic studies show that PARP inhibition is one mechanism by which therapeutic amelioration is achieved. In total, this manuscript uncovers new potential treatments for RP suitable for future study, and also outlines a powerful approach for drug discovery that could be applied to other rare diseases with available pre-clinical models.

**Decision letter after peer review:**

Thank you for submitting your article "Large-scale phenotypic drug screen identifies neuroprotectants in zebrafish and mouse models of retinitis pigmentosa" for consideration by *eLife*. Your article has been reviewed by 3 peer reviewers, including James J Dowling as the Reviewing Editor and Reviewer #1, and the evaluation has been overseen by Didier Stainier as the Senior Editor. The following individual involved in review of your submission has agreed to reveal their identity: Elise Heon (Reviewer #3).

The reviewers have discussed the reviews with one another and the Reviewing Editor has drafted this decision to help you prepare a revised submission.

As the editors have judged that your manuscript is of interest, but as described below that substantial additional experiments are required before it is published, we would like to draw your attention to changes in our revision policy that we have made in response to COVID-19 (https://elifesciences.org/articles/57162). First, because many researchers have temporarily lost access to the labs, we will give authors as much time as they need to submit revised manuscripts. We are also offering, if you choose, to post the manuscript to bioRxiv (if it is not already there) along with this decision letter and a formal designation that the manuscript is "in revision at *eLife*". Please let us know if you would like to pursue this option. (If your work is more suitable for medRxiv, you will need to post the preprint yourself, as the mechanisms for us to do so are still in development.)

Summary:

Zhang et al. describe a novel multi organism drug discovery pipeline for rod photoreceptor cell survival. The goal of the study is to identify new potential therapeutics for RP and IRD. They identify several hits using a zebrafish model, and then test these hits in murine cell and organoid models. Finally, they test one drug in the eye of a mouse model.

Essential revisions:

There was general enthusiasm among the reviewers for the overall concept and for the pipeline. However, the study was felt to be incomplete and lacking essential data, as well as to lack important validation of key assumptions. This was particularly viewed to be important because the general zebrafish screening methodology has already been published by the group. There are two overarching areas where the manuscript was felt to be preliminary.

The first relates to the drug development aspect of the story. No single hits proved effective across all models. Intriguingly, combinatorial treatments appeared to be more effective in the zebrafish model. However, these were not tested in the murine models. This was viewed by all reviewers as an essential experiment for the overall significance and impact of the study. In addition, the combinatorial impact as graded above single drug alone should be examined (as in, instead of yes/no).

The second relates to the proposed mechanisms of action. There is a lack of convincing evidence to support the proposed mechanisms of therapeutic benefit. For example, the hypothesis related to protection from non-apoptotic cell death could be strengthened by biochemical assessment of cell death pathways as a means of showing that these pathways are truly modified or not. In addition, and specifically related to TDP1, there is an opportunity to use genetic modification to confirm inhibition of this protein (usually potentially four guide approach for F0 KO).

Mechanistic studies would improve the validity and present the opportunity to advance disease understand and enhance translatability.

1. Line 116: States the transgenic fish used in this study model the onset of retinitis pigmentosa (RP). There is no data supporting this claim. Is metronidazole /nitroreductase system activating the same cell death pathways activated in animals with RP? The references given for the disease indicate multiple possibilities including cGMP, PARP-1, Calpain, etc. Does the Ntr/metronidazole system engage these cell death pathways? Furthermore, are young rods the same as more mature rods? Better justification of the model is needed or the model should be changed.

2. What is the effect of candidates on YFP stability and rhodopsin promoter activity?

3. Line 150-152: States that previously identified compounds (17) that showed retinal neuroprotection had no effect on the transgenic zebrafish assay used in this study. This is very concerning, perhaps invalidating the whole assay and some discussion is warranted.

4. The tenuous connection to parthanatos and necroptosis in their mechanisms of action (MOA) analysis appears to be the weakest part of this paper. Why not explore this further and actually do some parthanatos and necroptosis analysis with the already identified compounds?

5. Figure 4: It is surprising that inhibiting apoptosis did not result in increased YFP expression when the transgenic system used is known to stimulate apoptosis. What does this mean? It is important to show that the apoptosis inhibitor used actually inhibited apoptosis under the conditions used.

6. The mouse models used appear haphazardly put together. Wildtype, Rd1, and Rd10 of different backgrounds were all used individually in three different tests that were ultimately compiled to screen the same compounds. The genetic backgrounds of the mice, as the authors themselves have mentioned, becomes a significant confound. It would be better if they either stuck with one model or used both Rd1 and Rd10 in all of their experiments. They attempted to link it back to the mechanism of action results and to the different cell death pathways, but this is not very convincing without actual data.

7. There is mention of a mystery positive control for which no data is shown Some information would be useful to best interpret their results from the single candidates and the combinatorial experiments.

---

## [Author Response]

Essential revisions:There was general enthusiasm among the reviewers for the overall concept and for the pipeline. However, the study was felt to be incomplete and lacking essential data, as well as to lack important validation of key assumptions. This was particularly viewed to be important because the general zebrafish screening methodology has already been published by the group. There are two overarching areas where the manuscript was felt to be preliminary.The first relates to the drug development aspect of the story. No single hits proved effective across all models. Intriguingly, combinatorial treatments appeared to be more effective in the zebrafish model. However, these were not tested in the murine models. This was viewed by all reviewers as an essential experiment for the overall significance and impact of the study.

We wish to clarify that three compounds were effective in zebrafish, mouse primary retinal cell cultures and *rd1* retinal explant models. Unfortunately, due to budgetary constraints, we were only able to test a single compound in the *rd10* mouse model in vivo, a long-release formulation of DHA (PLGA-DHA). Unfortunately, this test was not successful. However, there a potential technical explanations for this result which we now discuss in more detail. Namely, we posit that the absence of DMSO in the injection may have limited bioavailability (based on in vitro release kinetics requiring DMSO; see lines 414-416 and 421-423). We plan to address this issue in follow up studies.

For the requested combinatorial drug tests, pairs of all eleven lead compounds were assayed for enhanced survival effects using mouse primary photoreceptor cell culture models using both stressor compounds, thapsigargin and tunicamycin – a total of 136 new assays. The results showed a trend toward enhanced survival effects for 30 of 55 pairs for the thapsigargin-induced cell death assay and for 20 of 55 pairs for the tunicamycin-induced cell death assay (new Figure 4—figure supplement 2). However, unlike zebrafish assays, no wholly additive effects were evident (i.e., survival effects ≥ to the sum of the individual effects of both test compound). In addition, the enhanced survival effects observed for paired compounds did not reach statistical significance relative to single compound controls after corrections for multiple comparisons. These new findings are presented at lines 535-544 and in Figure 4—figure supplement 2. We posit that additive effects may require in vivo context, such as interactions with other cell types or tissues, highlighting a potential advantage of whole organism screening. This is now discussed at lines 722-728.

In addition, the combinatorial impact as graded above single drug alone should be examined (as in, instead of yes/no).

In a new summary Figure 11, section C, we provide the quantified survival effects of each individual compound (blue boxes), each compound pair (grey boxed, no additive effect;, pink boxes, additive effects; red boxes, synergistic effects) and, for easy comparison, the summed value of the two individual compound controls (in parenthesis). Equivalent data is provided for the mouse primary retinal cell culture assays (Figure 4—figure supplement 2C-D)

The second relates to the proposed mechanisms of action. There is a lack of convincing evidence to support the proposed mechanisms of therapeutic benefit. For example, the hypothesis related to protection from non-apoptotic cell death could be strengthened by biochemical assessment of cell death pathways as a means of showing that these pathways are truly modified or not. In addition, and specifically related to TDP1, there is an opportunity to use genetic modification to confirm inhibition of this protein (usually potentially four guide approach for F0 KO).

The following new experiments were performed in our zebrafish rod cell ablation model to more thoroughly interrogate the mechanism NTR/Mtz-mediated cell death and the MOA of lead compound survival effects:

1. Assessment of rod cell survival effects resultant to CRISPR/Cas9-based knockdown of *parp1*, *ripkl1*, *casp3a/3b*, and *tdp1* – i.e., genes required for PARP-dependent cell death (parthanatos and/or cGMP-dependent cell death), necroptosis, and apoptosis, respectively, as well as *tdp1* for a DNA repair signaling control. The four sgRNA approach was used, as suggested. These new findings are presented at lines 483-494 and shown in a newly added figure (Figure 8A)

2. Control experiment testing effects of CRISPR/Cas9-based knockdown of the same four gene targets on rod cell development, presented at lines 494-499 (Figure 8B).

3. Confocal in vivo imaging of whole zebrafish retinas for conditions tested in Figure 7A and 7B, presented at lines 499-501 (Figure 8—figure supplement 1A).

4. qPCR confirmation of target gene knockdown, presented at line 501-502 (Figure 8—figure supplement 1B, Supplementary File 3)

5. Western blots and quantification of PAR accumulation (as a measure of PARP activation) following Mtz treatment of NTR-expressing transgenic fish, presented at lines 504-514 (Figure 8—figure supplement 1C-D).

6. Assessment of neuroprotective effects of five lead drugs (and a PARP inhibitor control) following CRISPR/Cas9-based knockdown of *parp1* to test if enhanced survival requires parp1 inhibition.

These new findings are presented at lines 515-529 and shown in a newly added figure (Figure 9).

The results support the hypothesis that non-apoptotic pathways, namely PARP-dependent cell death, and to a lesser degree necroptosis, mediate NTR/Mtz-induced ablation of rod photoreceptors. We also confirmed that the neuroprotective effects of two of four lead compounds linked to PARP inhibition were PARP dependent – controls confirmed PARP dependence of a known inhibitor (BMN) and PARP independence of a lead not linked to PARP (WAR). The results are briefly summarized below:

1. Knockdown of *parp1* significantly improved rod cell survival while *ripk1l* knockdown had a marginal survival effect. Conversely, *casp3a/3b* and *tdp1* gene knockdown showed no neuroprotective effect. These results are consistent with chemical inhibitor assays (Figure 7) and support the hypothesis that PARP signaling plays a role NTR/Mtz-mediated rod cell death.

a. Note, the negative *tdp1* result could be due to less efficient knockdown (Figure 8— figure supplement 1B). However, it is also possible the marginal neuroprotective effects of two of three Tdp1 inhibitors (Figure 7) were mediated indirectly through PARP – consistent with the results of a comprehensive quantitative HTS screen of 400,000 compounds designed to identify Tdp1 inhibitors, resulted in the isolation of five hits that were actually PARP inhibitors rather than Tdp1 inhibitors (Murai et al., 2014).

2. Knockdown of *casp3a/3b*, but no other targets, resulted in increased rod neogenesis, consistent with inhibition of developmental apoptosis (and confirming efficacy of *casp3a/3d* knockdown).

3. PAR accumulated in Mtz-treated *rho:YFP-NTR* fish, i.e., our inducible rod cell ablation model. The accumulation of PAR polymer confirms PARP activation in NTR-expressing fish exposed to Mtz prodrug.

4. Upon *parp1* knockdown, the neuroprotective effects of a known PARP inhibitor (control) were lost, as expected. In addition, two of four lead compounds predicted to be PARP inhibitors also lost neuroprotective activity, while a compound not linked to PARP maintained neuroprotective activity (see new Figure 9). These results suggest the neuroprotective effects of a subset of lead compounds are PARP-dependent, as suggested by HTS studies curated at PubChem.

In summary, chemical inhibitor assays (Figure 7), gene knockdowns (Figure 8A, Figure 8—figure supplement 1A), PAR accumulation analysis (Figure 8—figure supplement 1C-D) and drug testing in *parp1* knockdown larvae (Figure 9) are all consistent, suggesting the NTR/Mtz-induced rod cell ablation is mediated in large part by PARP-dependent cell death. PARP is activated in response to DNA damage, with extensive DNA damage leading to PARP-dependent cell death. This is also consistent with the proposed mechanism of NTR-mediated cell death; NTR-based reduction of Mtz leading to the production of cytotoxic hydroxylamine derivatives, a process that either directly or indirectly leads to DNA damage, and subsequent DNA damage-induced cell death. That PARP-dependent cell death is now strongly implicated in Parkinson’s disease and retinitis pigmentosa is discussed regarding the potential for the NTR/Mtz system to function as an inducible neurodegenerative disease modeling system, lines 131-135 and 693-695.

Mechanistic studies would improve the validity and present the opportunity to advance disease understand and enhance translatability.1. Line 116: States the transgenic fish used in this study model the onset of retinitis pigmentosa (RP). There is no data supporting this claim. Is metronidazole /nitroreductase system activating the same cell death pathways activated in animals with RP? The references given for the disease indicate multiple possibilities including cGMP, PARP-1, Calpain, etc. Does the Ntr/metronidazole system engage these cell death pathways?

Multiple types of cell death have been associated with RP – including apoptosis and PARPdependent pathways – but the molecular etiology of the disease remains unclear. At the cellular level, RP initiates with the loss and/or functional compromise of rod photoreceptors. The Mtz/NTR-induced rod cell ablation line is thus able to physiologically mimic this key feature of RP. In addition, as noted in the Discussion section, *lines 448-451 of the original submission*, “Prior reports had suggested the NTR/Mtz system elicits caspase-3 activation and apoptotic cell death (Chen et al., 2011). Initially, we had used this as a rational for pursuing a neuroprotective screen with the *rho:YFP-NTR* line.” In keeping with that, and assuming that rod cells die by apoptosis in RP and the NTR/Mtz system, we referred to the *rho:YFP-NTR* line as an inducible RP model. We now include that rationale in the Introduction (lines 166-170). In the results, we also initially refer to the *rho:YFP-NTR* line as “enabling targeted ablation of rod photoreceptors” (line 147) and “physiologically mimicking the onset of RP” (line 156) rather than as model of RP. Our intent was to delay discussion of the line as a model of RP (except the Abstract, simply to be succinct) until that possibility was clarified further during analysis of cell death mechanisms elicited by the NTR/Mtz system.

Truth be told, at the outset of our study we too had concerns about the disease relevance of the NTR/Mtz system, positing that a chemical screen may only serve to identify inhibitors of NTR. Thankfully, the results of our screen lent credence to the idea that NTR/Mtz system may prove useful as an inducible neurodegenerative disease modeling approach. However, the rationale is not what we initially assumed, i.e., that NTR/Mtz and RP both incur apoptotic cell death. Rather, our prior and newly generated data strongly suggest that apoptosis is not the primary mechanism by which the NTR/Mtz system mediated cell loss occurs, at least not in rod cells. Chemical inhibitor assays, gene knockdowns, Western blots and compound tests in *parp1* knockdown larvae all suggest NTR/Mtz-mediated rod cell ablation is mediated predominantly by a PARP-dependent cell death mechanism (see response to Essential revision 2, above). We were intrigued by recent findings from the Dawson lab that strongly implicate PARP dependent parthanatos in Parkinson’s disease (Kam et al., 2018) Accordingly, we worked with them to expand our analysis of PARP-dependent signaling in the *rho:YFP-NTR* line. This also motivated us to investigate the mechanisms of cell death implicated in RP more thoroughly.

The predominant study cited in support of apoptosis in RP models was published over 25 years ago (Portera-Cailliau et al., 1994). These investigators assessed apoptotic cell death in the Pde6b^rd1^ (formerly rd/rd), Prph^Rd2^ (formerly rds/rds) and Q344ter models of RP using the TUNEL assay and gel electrophoresis for internucleosomal DNA fragmentation. The data are consistent with increased apoptosis in the photoreceptor layer in all models, however, quantified data was not statistically evaluated. No other cell death pathways were investigated since this issue was largely viewed as a binary question at the time: apoptosis versus necrosis. The cell death pathway landscape has expanded considerably since then (Galluzzi et al., 2018). Using a comprehensive approach to evaluate multiple cell death mechanisms across ten mammalian models of photoreceptor degeneration, Arango-Gonzalez and colleagues found non-apoptotic mechanisms predominated in all models, whereas concordant evidence for apoptosis was limited to the S334ter model of autosomal dominant RP (Arango-Gonzalez et al., 2014). Interestingly, given our results, PARP activation was upregulated in all seven models of RP tested by Arango-Gonzalez and colleagues.

Combined with our expanded analysis of cell death mechanisms in the *rho:YFP-NTR* line, and in terms of the MOA of lead compounds, we speculate that the NTR/Mtz system may be useful as an inducible model of neurodegenerative diseases for which PARP-dependent cell death is implicated (e.g., Parkinson’s disease, RP). All disease models are imperfect to one degree or another, and caveats continue to attend the use of the NTR/Mtz system for this purpose. However, we feel the potential for expanded utility of the NTR/Mtz system as an inducible model for PARP-dependent cell death warrants discussion here. New results are presented and discussed as referenced in response to Essential revision 2, above.

Furthermore, are young rods the same as more mature rods? Better justification of the model is needed or the model should be changed.

Young rod photoreceptors have developed key features of mature rods, e.g. rod outer segments, synapses and rhodopsin expression as early as 3 dpf (Schmitt and Dowling, 1999). Moreover, low levels of rod function can be detected by ERG at 5 dpf (Moyano et al., 2013) and rod-driven behavioral responses can be assessed by 6 dpf (Venkatraman et al., 2020). Larval stage zebrafish are amenable to the large-scale screening approach that anchors our study, whereas juvenile and adult fish cannot be used for screens at this scale. Therefore, we opted to conduct our screen in larvae at 5-7 dpf.

Clearly, it will be important to test whether our lead compounds are effective in adult fish as well. However, we ask that the reviewers allow us to investigate that question in follow on studies.

Our use of multiple RP models was used to identify potential “pan-RP” therapeutics as well as to circumvent the limitations of any specific model. As discussed above, we too had concerns about using *rho:YFP-NTR* fish for the primary screen, despite initial assumptions that apoptosis was commonality to RP and the NTR system. Serendipitously, the results of the screen proved insightful in terms of clarifying cell death mechanisms at play on both RP and the NTR system.

Finally, changing the model is simply not be tenable as it would require the entire project to be reinitiated. Given the links between PARP-dependent cell death in the NTR/Mtz system and genetic models of RP discussed above, we feel that the model choice has been adequately justified. That these insights were generated largely in hindsight, during pursuit of lead compound MOA, also highlights the value of agnostic screening approaches for revealing new insights.

2. What is the effect of candidates on YFP stability and rhodopsin promoter activity?

We performed a test in which *rho:YFP-NTR* larvae were treated with lead compounds for two days (from 5-7 dpf) to assess effects on rod cell neogenesis and YFP fluorescence in the absence of Mtzinduced rod cell ablation. No leads showed significantly increased YFP signals compared to controls (former Supplementary Figure 4, now Figure 2—figure supplement 2). This result indicated that lead compounds did not enhance YFP stability or stimulate rhodopsin promoter activity (see lines 296-305).

3. Line 150-152: States that previously identified compounds (17) that showed retinal neuroprotection had no effect on the transgenic zebrafish assay used in this study. This is very concerning, perhaps invalidating the whole assay and some discussion is warranted.

In an effort to identify a positive control compound for our large-scale screen, we tested 17 compounds previously reported as being neuroprotective in mammalian RP models. Unfortunately, none promoted rod cell survival in *rho:YFP-NTR* fish. However, neither do these compounds promote rod cell survival in human RP patients. The cross-species validity of these compounds is therefore questionable. To our knowledge, no compound has been reported to be neuroprotective across fish and mammalian RP models, nor between mammalian models and human patients. The point of our screen was to fill this gap in the hopes of identifying neuroprotective therapeutics that work across species, thus potentially by highly conserved MOA. Serendipitously, an unpublished compound recently identified as a neuroprotectant in mouse retinal disease models by the Zack lab proved effective in promoting enhanced YFP signals in our system and was therefore used as the positive control. We have expanded the discussion of these points in the revised manuscript (see lines 206-214).

4. The tenuous connection to parthanatos and necroptosis in their mechanisms of action (MOA) analysis appears to be the weakest part of this paper. Why not explore this further and actually do some parthanatos and necroptosis analysis with the already identified compounds?

To strengthen the MOA analysis, we performed gene knockdown experiments, a PAR accumulation analysis, and combined gene knockdown with tests of lead compound efficacy (see response to Essential revision 2 and Major Point 1, above). The results implicate PARP-dependent cell death (parthanatos and/or cGMP-dependent cell death) and, to a lesser extent, necroptosis in mediating Mtzinduced cell death.

5. Figure 4: It is surprising that inhibiting apoptosis did not result in increased YFP expression when the transgenic system used is known to stimulate apoptosis. What does this mean? It is important to show that the apoptosis inhibitor used actually inhibited apoptosis under the conditions used.

Unfortunately, the apoptosis inhibitor we used initially (belnacasan, BEL) was selected in error. It is actually an inhibitor of caspases 1 and 4, thus a pyroptosis inhibitor. In the revised manuscript we have corrected this error by testing three inhibitors of caspases 3 and 9 (Ac-DEVD-CHO; CASI, caspase3/7 inhibitor I; CASVII, caspase 3 inhibitor VII). Relative activity of these reagents against caspase enzyme variants is shown in Table 2. In addition, we added gene knockdown experiments cotargeting *casp3a/3b*. The results summarized in response to Essential revision 2 and Major Point 1 (above) strongly suggest that NTR/Mtz-induced DNA damage leads to DNA damage-induced cell death – that is, PARPdependent parthanatos.

More broadly, and as our lab developed the NTR-based cell ablation system in zebrafish, we feel it is important that we further clarify this issue. The evidence for NTR/Mtz-induced apoptosis has relied largely on Terminal deoxynucleotidyl transferase dUPT nick-end labeling (TUNEL). This assay detects the presence of fragmented DNA. DNA fragmentation is associated with apoptosis, specifically caspaseactivated DNase activity, but TUNEL is not a definitive marker of apoptosis, requiring a secondary assay to confirm the mechanism of cell death (Baima and Sticherling, 2002; Kanoh et al., 1999; Lawrence et al., 2013; Loo, 2011). Moreover, the NTR/Mtz system incurs DNA damage directly. Thus, TUNEL can only be used to confirm NTR/Mtz-induced DNA damage, not to determine the nature of the cell death pathways involved. Complementary evidence of activated caspase-3 in Mtz-treated NTR-expressing cells has largely relied on images showing limited co-labeling (Godoy et al., 2015; Wang et al., 2004). Where qPCR data showing quantitative increases in caspase 3 and caspase 7 have been shown for the NTR/Mtz system, total RNA from larvae was used (Lei et al., 2017). Given that phagocytic macrophages undergo apoptosis (Murray and Wynn, 2011) and that macrophages clear NTR/Mtz-ablated cells (White et al., 2017), the qPCR assay cannot account for which cell types produce elevated caspase levels. Of note, our lab has struggled to demonstrate statistically validated increases in markers of apoptosis in Mtz-treated, NTR-expressing cells (e.g., activated caspase-3, secreted annexin-V). Finally, the use of Akt1, Stat3, and EPV E6 overexpression to protect cells from NTR/Mtz-induced “apoptosis” (Chen et al., 2011) is also in question. It is now known that Akt1 and Stat3 protect against anoikis, a specific form of “intrinsic apoptosis” triggered by loss of integrin-dependent anchorage to the extracellular matrix, thus relevant only to anchorage-dependent cells (e.g., epithelia). Relevance of anoikis to NTR/Mtz-induced ablation of non-epithelial cells is unclear. EPV E6 can inhibit cell death through the degradation of p53 (Scheffner et al., 1990). However, this is not necessarily indicative of inhibition of apoptosis as p53 is now known to be involved in autophagic cell death, ferroptosis, and necroptosis in addition to classical apoptosis, nonclassical apoptosis (Yamada and Yoshida, 2019). Moreover, E6 also targets the apoptosis inducing factor (AIF) for degradation. Despite the name, AIF is also required for PARP-dependent cell death (Wang et al., 2011). Given these caveats, levied largely by recent expansions in understanding of the intricacies of regulated cell death (Galluzzi et al., 2018), the mechanism by which the NTR/Mtz system elicits cell death can no longer be assumed to be via apoptosis and clearly warrants further investigation. We hope that the cell death pathway analysis we performed in pursuit of clarifying the MOA of lead compounds (Figures 7-9, Figure 7—figure supplement 1 and Figure 8—figure supplement 1; see also Response to ER2 and Major Point 1) will help to spur further investigations of NTR/Mtz-mediated cell death in other cellular targets.

6. The mouse models used appear haphazardly put together. Wildtype, Rd1, and Rd10 of different backgrounds were all used individually in three different tests that were ultimately compiled to screen the same compounds. The genetic backgrounds of the mice, as the authors themselves have mentioned, becomes a significant confound. It would be better if they either stuck with one model or used both Rd1 and Rd10 in all of their experiments. They attempted to link it back to the mechanism of action results and to the different cell death pathways, but this is not very convincing without actual data.

The use of multiple RP models across species was by design. The intent of our screening cascade was to identify mutation-independent therapeutics directed toward conserved etiologies; this was in fact the directive of our funding agency, the Foundation Fighting Blindness. We apologize that our rationale was not adequately emphasized in the initial manuscript and have revised the Abstract (lines 4346) and Introduction (lines 75-77) to correct this oversight.

In addition, as part of the revision process we opted to simplify the mouse cell culture analysis by reporting only survival effects on primary photoreceptor cells (rather than all retinal neurons). All eleven lead compounds were tested alone and in pairs using two stressor-induced photoreceptor cell death models. Nine of eleven lead compounds were found to be effective in at least one of these assays (Figure 4, Figure 8—figure supplement 2).

The rationale for using the *rd1* model for retinal explant tests was further elaborated (lines 380-387). The rationale for using the *rd10* model for in vivo testing in mice was provided previously (lines 404-408 and 614-616) and rationale for cross-species testing was expanded to “cross-species/multimodel” testing to emphasize this more strongly (abstract lines 57-59). The data linking *rd1* and *rd10* mouse models of RP to PARP-dependent cell death were provided (Arango-Gonzalez et al., 2014; Power et al., 2019).

7. There is mention of a mystery positive control for which no data is shown Some information would be useful to best interpret their results from the single candidates and the combinatorial experiments.

The positive control used for the large-scale drug screen is a tyrosine kinase inhibitor identified as a neuroprotectant by the Zack lab. A manuscript describing the structure and function of this compound in retinal disease contexts is in preparation. For our purposes, it was used solely as a means of assessing the integrity of weekly assay runs during screening and subsequent validation testing. We ask that the Zack lab be allowed to present the structure and function of this discovery independently as the screening process underlying that discovery represents an extensive amount of work for the lead author of that study. Once their study has been published, we will provide an update that clarifies the identity of the positive control.